# DENR controls JAK2 translation to induce PD-L1 expression for tumor immune evasion

Baiwen Chen[1,2,7], Jiajia Hu[3,7], Xianting Hu[1,7], Huifang Chen[2], Rujuan Bao[2], Yatao Zhou[2], Youqiong Ye [2], Meixiao Zhan[4✉], Wei Cai[5✉], Huabin Li[1✉] & Hua-Bing Li [2,5,6✉]

RNA-binding proteins (RBPs) can recognize thousands of RNAs that help to maintain cell homeostasis, and RBP dysfunction is frequently observed in various cancers. However, whether specific RBPs are involved in tumor immune evasion by regulating programmed death ligand-1 (PD-L1) is unclear. Here, we perform targeted RBP CRISPR/Cas9 screening and identify density regulated re-initiation and release factor (DENR) as a PD-L1 regulator. DENR-depleted cancer cells exhibit reduced PD-L1 expression in vitro and in vivo. DENR depletion significantly suppresses tumor growth and enhances the tumor-killing activity of CD8+ T cells. Mechanistically, DENR antagonizes the translational repression of three consecutive upstream open reading frames (uORFs) upstream of Janus kinase 2 (*Jak2*); thus, DENR deficiency impairs JAK2 translation and the IFNγ-JAK-STAT signaling pathway, resulting in reduced PD-L1 expression in tumors. Overall, we discover an RBP DENR that could regulate PD-L1 expression for tumor immune evasion, and highlight the potential of DENR as a therapeutic target for immunotherapy.

[1] ENT institute and Department of Otorhinolaryngology, Eye & ENT Hospital, Fudan University, Shanghai 200031, China. [2] Shanghai Institute of Immunology, State Key Laboratory of Oncogenes and Related Genes, Shanghai Jiao Tong University School of Medicine, Shanghai 200025, China. [3] Department of Nuclear Medicine, Ruijin Hospital, Shanghai Jiao Tong University School of Medicine, Shanghai 200025, China. [4] Zhuhai Interventional Medical Center, Zhuhai Precision Medical Center, Zhuhai People's Hospital, Zhuhai Hospital of Jinan University, Zhuhai, Guangdong 519000, China. [5] Department of General Surgery, Shanghai Minimally Invasive Surgery Center, Shanghai Institute of Immunology, Ruijin Hospital, Shanghai Jiao Tong University School of Medicine, Shanghai, China. [6] Shanghai Jiao Tong University School of Medicine - Yale Institute for Immune Metabolism, Shanghai Jiao Tong University School of Medicine, Shanghai 200025, China. [7] These authors contributed equally: Baiwen Chen, Jiajia Hu, Xianting Hu. ✉email: zhanmeixiao1987@126.com; caiwei@shsmu.edu.cn; allergyli@163.com; huabing.li@shsmu.edu.cn

Programmed death ligand 1 (PD-L1) is typically expressed on tumor cells and immune cells such as macrophages and dendritic cells, and the interaction between PD-L1 and its receptor, programmed cell death-1 (PD-1), on T cells inhibits the tumor-killing activity of these cells[1–3]. One of the most exciting progressions in cancer research in recent years has been the development of the PD1/PD-L1 blockade, which benefits patients with diverse types of cancers[4–6]. However, only a small proportion of patients can benefit from anti-PD-1/PD-L1 therapy[7]. Those tumors with high PD-L1 expression are with poor prognosis but considered to be more sensitive to PD-1/PD-L1 blockade[8]. Nevertheless, the mechanism that regulates PD-L1 expression in tumor cells has not been fully elucidated, understanding of which regulatory network would aid to develop potential therapeutic strategies to enhance efficacy of anti-tumor immunotherapy.

PD-L1 expression can be induced by inflammatory factors such as interferon (IFN)γ, IFNα, IFNβ, tumor necrosis factor (TNF)α, and transforming growth factor (TGF)β. IFNγ, which is generally produced by T cells, is considered to be the most prominent inducer of PD-L1[9]. The binding of IFNγ to its receptors activates the JAK-STAT signaling pathway, thereby up-regulating the expression of a series of genes, including *PD-L1*[9]. Besides inflammatory signaling, the elevated expression of oncogenic transcription factors, such as MYC[10], HIF1α[11], NF-κB[12], and BRD4[13], in cancers has also been demonstrated to directly regulate PD-L1 transcription. In addition, increasing evidence demonstrated that PD-L1 also undergoes different posttranscriptional regulation. CKLF-like MARVEL transmembrane domain-containing 6 has been shown to positively regulate PD-L1 by direct binding to increase its half-life[14,15], and miR-513[16] and miR-155[17] suppress PD-L1 expression by directly binding to the 3′ UTR of *PD-L1* mRNA.

RNA-binding proteins (RBPs) are involved in all aspects of RNA metabolism, including splicing, exportation, location, modification, translation, and decay[18]. Although PD-L1 regulation has been extensively studied at the transcriptional and posttranscriptional levels, whether a particular RBP is involved in the PD-L1 regulation networks is still unknown. To systematically discover the involvement of the RBP regulatory pathways in PD-L1 expression, we perform targeted pooled RBP CRISPR/Cas9 screening and identify an RBP, density-regulated re-initiation and release factor (DENR), as a PD-L1 regulator. We find three upstream open reading frames (uORFs) in the Janus kinase 2 (*Jak2*) 5′ UTR region that constitutively repress the translation of JAK2. DENR binds to 40S ribosomes and bypasses the three uORFs to reinitiate the translation of the *Jak2* main ORF. A deficiency in DENR leads to decreased JAK2 translation and impairs JAK-STAT signaling, thus reducing PD-L1 expression in tumor cells and repressing tumor growth by enhancing the anti-tumor killing activity of CD8[+] T cells in the tumor microenvironments.

## Results

**Pooled CRISPR screening identified DENR as a regulator of PD-L1.** Previously, we successfully performed the pooled RBP CRISPR screening of macrophages and identified m[6]A "writers" to be positive regulators of TNF expression and macrophage activation[19]. To unbiasedly and systematically screen for posttranscriptional regulators of PD-L1, we constructed a new customized RBP CRISPR mini-library with a larger RBP pool than our previous 782-RBP-gene CRISPR mini-library. The new version of the RBP library contained 10 single-guide (sg)RNAs per gene of 1467 known RBPs[20], as well as positive and negative control sgRNAs (Supplementary Data 1). We used fluorescence-activated single-cell sorting (FACS) to quantify the PD-L1 expression levels, which served as a reliable and convenient CRISPR screening readout.

IFNγ is generally considered to be the most potent inducer of PD-L1 expression[14], and murine macrophage-like RAW264.7 cells show a rapid response to IFNγ treatment, expressing high levels of surface PD-L1 (Supplementary Fig. 1a). We constructed a Cas9-expressing stable RAW264.7 cell line, then infected the cells with the above lentivirus RBP CRISPR library at a multiplicity of infection of 0.3 (Fig. 1a and Supplementary Fig. 1b) and selected them with puromycin for 7 days. Next, the cells were stimulated with IFNγ for 4 h to induce PD-L1 expression at a moderate level (Supplementary Fig. 1a), to allow easier identification of RBP genes that could either enhance or suppress PD-L1 expression as previously suggested. Then, the cells were sorted by FACS based on their surface expression of PD-L1 (Fig. 1b). Genomic DNA was extracted from PD-L1[High], PD-L1[Low], and unsorted cells, and the inserted sgRNAs were amplified and sequenced (Fig. 1b).

As expected, sgRNAs targeting known positive regulators of PD-L1, such as JAK1, JAK2, and STAT1 in the IFNγ signaling pathway[7], were identified in the PD-L1[Low] cells and no negative control sgRNAs were enriched (Fig. 1c). Among the top candidates in the PD-L1[low] cells, three components of the SWI/SNF chromatin remodeling complex, BRD2, BRD9, and SMARCD1, were identified (Fig. 1c). This complex has been reported to regulate the expression of PD-L1[21,22]. These findings demonstrated the validity of our screening setup.

To further validate our screening results, we tested the top hit genes enriched in PD-L1[low] cells by generating stable KO RAW264.7 cell lines using individual corresponding sgRNAs. The FACS results showed that Knocking-out those genes, *Denr*, *Brd2*, *Atp5sl*, *Kat7*, *Ndufa10*, all significantly reduced the expression of cell surface PD-L1 (Supplementary Fig. 1d). Noticeably, DENR knockout in Cas9-RAW264.7 cells could markedly reduce the total and surface PD-L1, which could be further validated by additional three specific sgRNAs (Fig. 1d, e). Collectively, we identified DENR to be a posttranscriptional regulator of PD-L1 by customized pooled RBP CRISPR screening.

**DENR regulates PD-L1 through the JAK-STAT signaling pathway.** DENR is reported to be an RBP that binds directly to transfer RNAs in vitro[23] and is involved in the translation re-initiation of target mRNAs with short 'start-stop' uORFs (e.g., AUGUGA)[24]. However, we did not find any classical conserved 'start-stop' ORF in the short 5′ UTR of the PD-L1 gene, suggesting indirect PD-L1 regulation by DENR. To test our hypothesis, we knocked out DENR gene in murine colon adenocarcinoma MC38 cells and assayed the protein and mRNA levels of PD-L1 by immunoblot, FACS and RT-qPCR respectively (Fig. 2a–c). In the absence of IFNγ, PD-L1 protein levels were almost undetectable (Fig. 2a), and there was no difference in mRNA levels between DENR KO and control cells (Fig. 2c). IFNγ treatment, in contrast, dramatically decreased both total and surface PD-L1 protein and mRNA levels in DENR KO cells compared to control cells (Fig. 2a–c), suggesting that DENR does not directly regulate PD-L1 translation but indirectly regulates it through IFNγ-induced transcriptional signaling pathways. In addition, we observed a reduction of PD-L1 after DENR KO in murine melanoma B16/F10 cells (Supplementary Fig. 2a, b), suggesting DENR's general role in reducing PD-L1 expression in different cell lines.

To identify the potential signaling pathway involved, we harvested IFNγ-treated control and DENR KO MC38 cells for RNA-seq. Bioinformatic analysis of the RNA-seq data found that

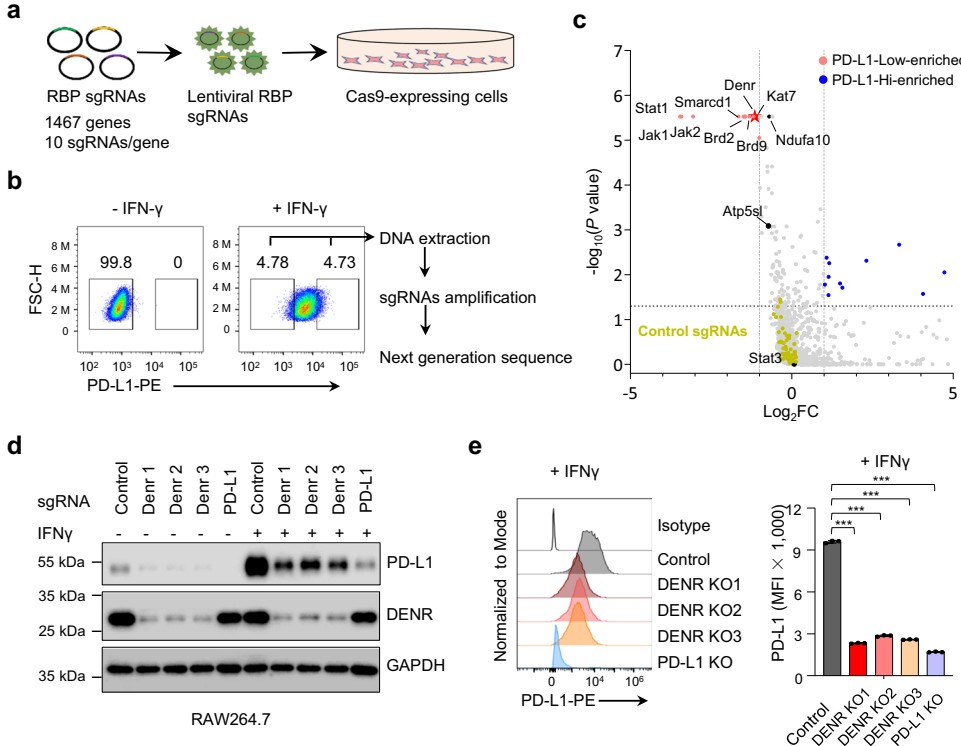

**Fig. 1 Identification of DENR as a PD-L1 regulator by CRISPR/Cas9 screening. a** Single-guide (sg)RNAs were packaged into a lentivirus and transfected into Cas9-expression RAW264.7 cells; cells were selected with puromycin (8 ng/ml) for 7 days. **b** RAW264.7 cells were incubated with 20 ng/ml IFN-γ for 4 h to induce PD-L1 at a moderate level; PD-L1$^{Low}$ and PD-L1$^{High}$ cells were sorted by FACS of IFNγ-treated cells, and the sgRNAs were amplified and sequenced. **c** Dots represent individual genes; red and blue dots indicate genes with significant enrichment within the PD-L1$^{Low}$ and PD-L1$^{High}$ populations, respectively; orange dots indicate negative control sgRNAs; black dots indicate *Stat3*, *Ndufa10*, and *Atps5l*. The experiment was repeated independently two times with similar results. **d** Western blotting validation of DENR and PD-L1 knockout with specific sgRNAs in RAW264.7 cells, ± IFNγ (20 ng/ml) for 24 h. The experiment was repeated independently three times with similar results. **e** Surface PD-L1 levels were analyzed by FACS in DENR and PD-L1 KO cells, + IFNγ (20 ng/ml) for 24 h ($n = 3$ independent samples). Results are representative of three biological replicates. Three technical replicates are shown. Data are presented as mean values ± SD. Two-sided, one-way ANOVA with Dunnett's post hoc test: *** $p < 0.001$. $p$ values from left to right: < 0.0001; < 0.0001; < 0.0001; < 0.0001. MFI median fluorescence intensity. Source data are provided as a Source Data file.

114 genes, including *PD-L1*, were significantly downregulated in DENR KO cells (Fig. 2d). KEGG analysis showed that these downregulated genes in DENR KO cells were enriched in JAK-STAT and PD-L1 signaling pathways (Fig. 2e), which could be verified by GESA analysis (Fig. 2f). In addition to PD-L1, the top down-regulated genes in DENR KO cells included *Cxcl10*, *Cxcl9*, *Cxcl1*, and *Irf1* (Fig. 2d), which are known to be specifically regulated by JAK-STAT signaling in response to IFNγ[25–27].

It has been thoroughly documented that IFNGR1 and IFNGR2 sense extracellular IFNγ, then transduce signals into the cell cytoplasm to activate JAK1 and JAK2 by phosphorylation[9,28]. The activated JAK1 and JAK2 subsequently phosphorylate STAT1 in most cells and STAT3 in some cells, and the pSTAT1/3 are translocated into the nucleus, where they bind to and promote the transcription of target genes such as *PD-L1*[9]. We thus hypothesized that DENR KO impaired the JAK-STAT signaling pathway. To test our hypothesis, we analyzed the protein and phosphorylation levels of STAT1 and STAT3 (Fig. 2g). Interestingly, we found the phosphorylation levels, but not the total protein levels, of STAT1 and STAT3 were dramatically reduced after IFNγ treatment, suggesting that the DENR-targeted gene was upstream of STAT1/3. Together, our data demonstrate that DENR regulates PD-L1 expression via the JAK-STAT signaling pathway.

**DENR directly regulates JAK2 expression.** To further explore the molecular mechanisms of DENR function in PD-L1 regulation, we determined the protein and phosphorylation levels of all

upstream factors of STAT1/3, including IFNGR1, IFNGR2, JAK1, and JAK2 (Fig. 3a and Supplementary Fig. 2c), by western blotting. We found that both the protein and phosphorylation levels, but not the mRNA levels of JAK2 were markedly reduced in DENR KO cells cultured with or without IFNγ treatment (Fig. 3a–c), while JAK1, IFNGR1, and IFNGR2 were unaffected by DENR depletion (Fig. 3a and Fig. 2d), indicating DENR's specific translational regulation over JAK2 expression. In addition, the reduction in JAK2 protein levels could be restored by the overexpression of an sgRNA-resistant DENR cDNA expression vector in DENR KO cells (Supplementary Fig. 2e), thus excluding the potential off-targeting effects of JAK2 by DENR sgRNAs. Furthermore, individually knocking out JAK1, JAK2, STAT1 showed a similar reduction of total and surface PD-L1 protein levels comparing to Knocking out DENR, but not STAT3 (Fig. 3d, e). These results were consistent with our CRISPR screening that JAK1, JAK2, STAT1 but not STAT3 sgRNAs were enriched in PD-L1$^{low}$ cells (Fig. 1c). Together, our data prove that DENR directly regulates JAK2 expression via translation, thus its downstream JAK2-STAT1 signaling pathway and then PD-L1 expression.

As an RBP, DENR likely regulates JAK2 at the post-transcriptional level, but it may also regulate JAK2 activity by protein-protein interactions, thus, we performed immunoprecipitation (IP) and found that DENR pulled down Multiple Copies In T-Cell Lymphoma-1 (MCTS1) (Supplementary Fig. 3a), a known DENR interaction partner protein. However, JAK2 was

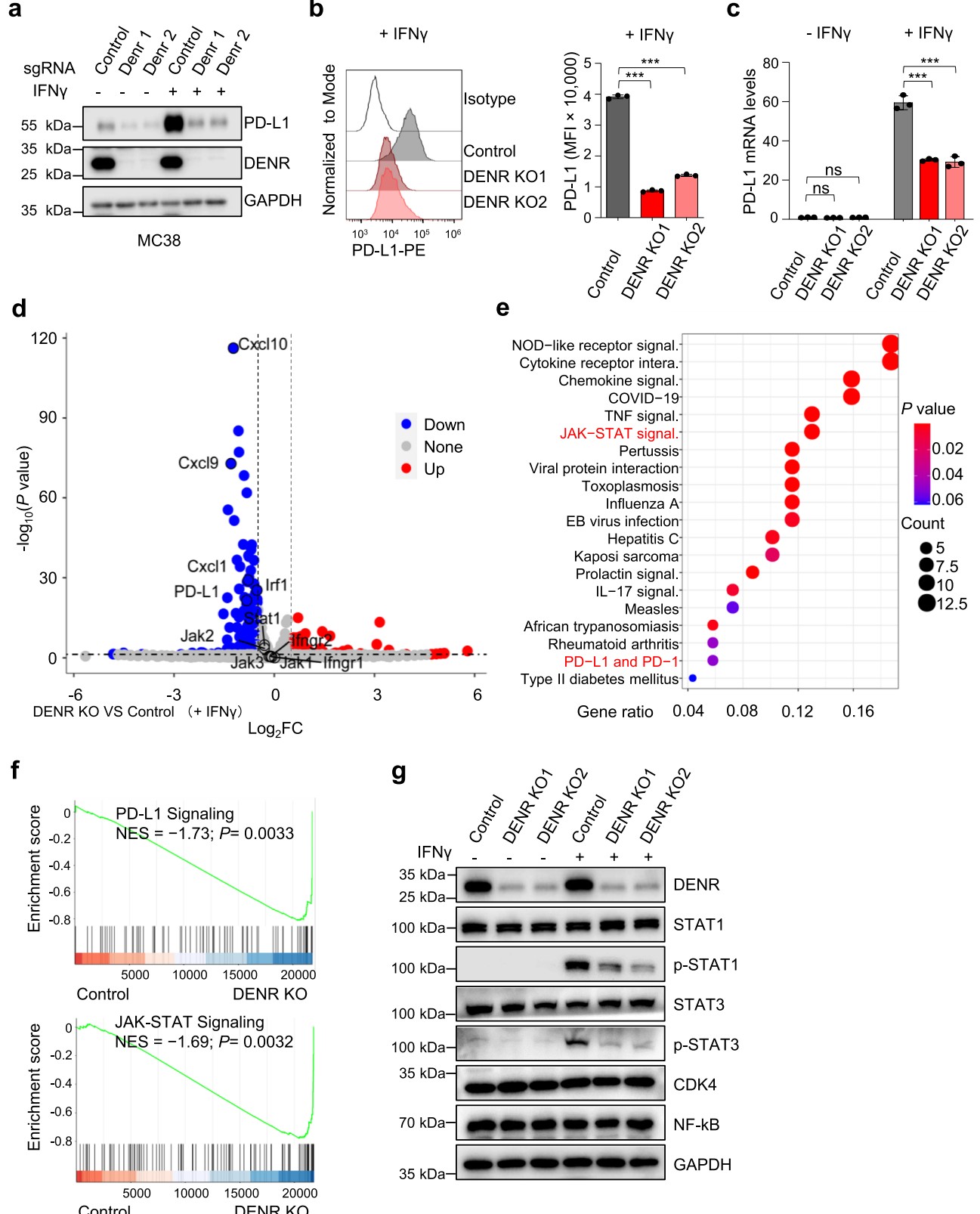

not detected as an immunoprecipitate with the anti-DENR antibody (Supplementary Fig. 3a). Reciprocally, JAK2 IP with the anti-JAK2 antibody pulled down STAT3 as expected, but not DENR (Supplementary Fig. 3a).

DENR is reported to regulate the translation of target mRNAs[29–31], and we found that the protein, but not the mRNA levels of JAK2 were reduced in DENR KO cells. In addition, we excluded the possibility of a protein–protein interaction between DENR and JAK2. All together, these data prompt us to hypothesize that DENR directly regulates the translation of JAK2, which in turn drives PD-L1 expression in response to IFNγ.

**Fig. 2 DENR depletion impaired the JAK-STAT signaling pathway. a** Western blotting validation of DENR knockout and PD-L1 expression with two specific sgRNAs in MC38 cells, ± IFNγ (20 ng/ml) for 24 h. The experiment was repeated independently three times with similar results. **b** Surface PD-L1 levels were analyzed by FACS in DENR and PD-L1 KO cells, + IFNγ (20 ng/ml) for 24 h ($n = 3$ independent samples). Results are representative of three biological replicates. Three technical replicates are shown. Data are presented as mean values ± SD. Two-sided, one-way ANOVA with Dunnett's post hoc test: *** $p < 0.001$. *P* values from left to right: <0.0001, <0.0001. **c** RT-qPCR analysis of PD-L1 mRNA levels normalized to β-actin mRNA in control and DENR KO MC38 cells, ± IFNγ (20 ng/ml) for 4 h ($n = 3$ biologically independent samples). Data are presented as mean values ± SD. Two-sided, one-way ANOVA with Dunnett's post hoc test: *** $p < 0.001$, ns: not significant. *P* values from left to right: 0.1059, > 0.9999, < 0.0001, < 0.0001. **d** Volcano plot of global gene expression in DENR KO versus control MC38 cells ($n = 2$ independent samples) upon IFNγ stimulation (20 ng/ml for 4 h). Upregulated genes are labeled as red dots, whereas downregulated genes are labeled in blue. **e** KEGG analysis of downregulated genes in DENR KO cells versus control cells following IFNγ treatment. **f** Gene Set Enrichment Analysis (GSEA) showing downregulation of JAK-STAT, and PD-L1 pathways in DENR KO cells. *p* value is calculated using GSEA Empirical phenotype-based permutation test, two-sided, and no adjustments were made for multiple comparisons. **g** Western blotting analysis of STAT1/3, p-STAT1/3, CDK4, and NF-kB expression in control or DENR KO MC38 cells, ± IFNγ (20 ng/ml) for 1 h. The experiment was repeated independently three times with similar results. Source data are provided as a Source Data file.

**uORFs inhibit the translation of JAK2 and are antagonized by DENR**. We next sought to understand how DENR regulates JAK2 translation. DENR has been reported to regulate the translation of target genes with uORFs[29,30]. Interestingly, we observed three conservative uORFs in both the murine and human JAK2 gene, with two copies of a short 'start-stop' uORF, AUGUGA, and a longer uORF, AUGUUCGA (Fig. 4a). Furthermore, knocking out DENR in human cancer cell lines, such as human colon HT29 cancer cells and human melanoma A375 cells, also resulted in reduced JAK2 protein levels (Supplementary Fig. 2d), indicating that the mechanism of DENR in regulating JAK2 expression is conservative in both murine and human.

To test the function of the uORFs upstream of the *Jak2* gene, we cloned the wildtype (WT) (three intact uORFs) or mutant (no uORF or only one longer uORF) *Jak2* 5′ UTRs into EGFP-HA-SV40 Poly(A) reporter constructs (Fig. 4b) which were transfected into WT B16/F10 cells, then measured the EGFP fluorescence by microscopy and the HA-tagged EGFP protein levels by western blot (Fig. 4c, d). The expression levels of the reports with three or one uORF were markedly repressed compared to that of the reporter lacking uORF, and the reporter with three uORFs was repressed more than the reporter with only one uORF (Fig. 4c, d). These data indicate that the uORFs upstream *Jak2* mRNA are indeed responsible for the repression of JAK2 translation efficiency.

Next, to determine whether the repressive function of the uORFs in *Jak2* mRNA is further regulated by DENR, reports were transfected into control and DENR KO cells (Fig. 4e). The proportion of EGFP-positive cells was reduced by up to 50% in DENR KO cells transfected with the reporters of three uORFs and by up to 15% in cells containing the report with one uORF, while reports with no uORF were not repressed by DENR depletion (Fig. 4f). In addition, the corresponding mRNA levels of EGFP reports were measured and no differences were observed in all experimental groups (Fig. 4g), suggesting that the difference in EGFP fluorescence is not due to EGFP transcription, but the translational regulatory effects of the upstream uORFs and associated DENR.

Furthermore, we analyzed the global genes and identified 2635 genes with at least 1 uORF. We then overlap those genes with translation deficiency genes defined from published ribosome profiling sequencing data set of DENR KO NIH3T3 cells (GSE116221)[32], the results showed there are 106 genes with at least 1 uORF and decreased translation efficiency, and *Jak2* was among them (Fig. 4h).

Together, these results show that the uORFs upstream *Jak2* mRNA inhibits the translation of JAK2, and the inhibitory effects of uORFs are directly antagonized by DENR.

**DENR deficiency stalls ribosomes between uORFs**. To explore the detailed molecular mechanisms of how DENR directly regulates JAK2 expression via upstream uORFs, we performed RNA immunoprecipitation with an anti-DENR antibody to detect DENR-bound RNAs (Fig. 5a). Unexpectedly, the qPCR amplification of the immunoprecipitated RNAs showed that DENR was bound to not only *Jak2* mRNAs, but also *Jak1* and *Gapdh* mRNAs lacking uORFs (Fig. 5b). Recent structural studies have shown that DENR binds to 40S ribosomes[33–35], thus we performed polysome fractionation analysis to determine whether DENR affects the translation efficiency of all or only certain mRNAs. Our results showed that DENR depletion did not affect the formation of global polyribosomes (Fig. 5c). In addition, we carried out RT-qPCR on each polysome fraction in Control and DENR KO cells, and the results showed that *Jak1* mRNAs without uORFs did not have any significant ribosome association changes with each fractions (Fig. 5d), however, the distribution of *Jak2* mRNAs displayed a leftward shift on the fractionation gradient (Fig. 5e), indicating that *Jak2* mRNAs associated with smaller polysomes upon DENR depletion, suggesting that the translation efficiency of JAK2 but not JAK1 were suppressed in DENR KO cells. Finally, we also reanalyzed a ribosome footprint sequencing data set (GSE140084)[30] in DENR KO versus WT HeLa cells and found that loss of DENR led to the accumulation of 40S and 80S on uORFs in *JAK2* (Fig. 5f) but not *JAk1* (Supplementary Fig. 3b). Together, our results suggest that DENR binds all mRNAs and bypass the uORFs when encountering them, thus promote translation efficiency of specific mRNAs with uORFs, specifically *Jak2* mRNAs.

Next, to further confirm that the global protein synthesis was not affected by DENR KO, we first performed Click-iT OPP Protein Synthesis assay and found no changes of the global incorporation efficiency of OP-puro analog into newly translated peptides in DENR KO over WT cells (Supplementary Fig. 3c, d). Then, we checked the phosphorylation status of translation initiation factors eIF2α, 4EBP1, and eEF2, whose phosphorylation would lead to protein synthesis inhibition[36,37]. The results showed that the phosphorylation status of eIF2α, 4EBP1, and eEF2 was un-altered in DENR KO cells (Supplementary Fig. 3e).

All together, these data exclude the possibility that DENR has a broad effect on the translation of all mRNAs and suggest that DENR constitutively associates with ribosomes, travels indiscriminately along all mRNAs, promotes translation re-initiation only when encountering uORFs belonging to certain mRNAs.

**DENR depletion attenuates PD-L1 expression and tumor growth in vivo**. So far, we have shown the molecular mechanism of how DENR regulates the translation of JAK2 to control the expression of PD-L1. Next, we sought to determine the in vivo role of DENR in tumor progression and PD-L1 expression. We first showed that DENR depletion did not affect tumor growth in

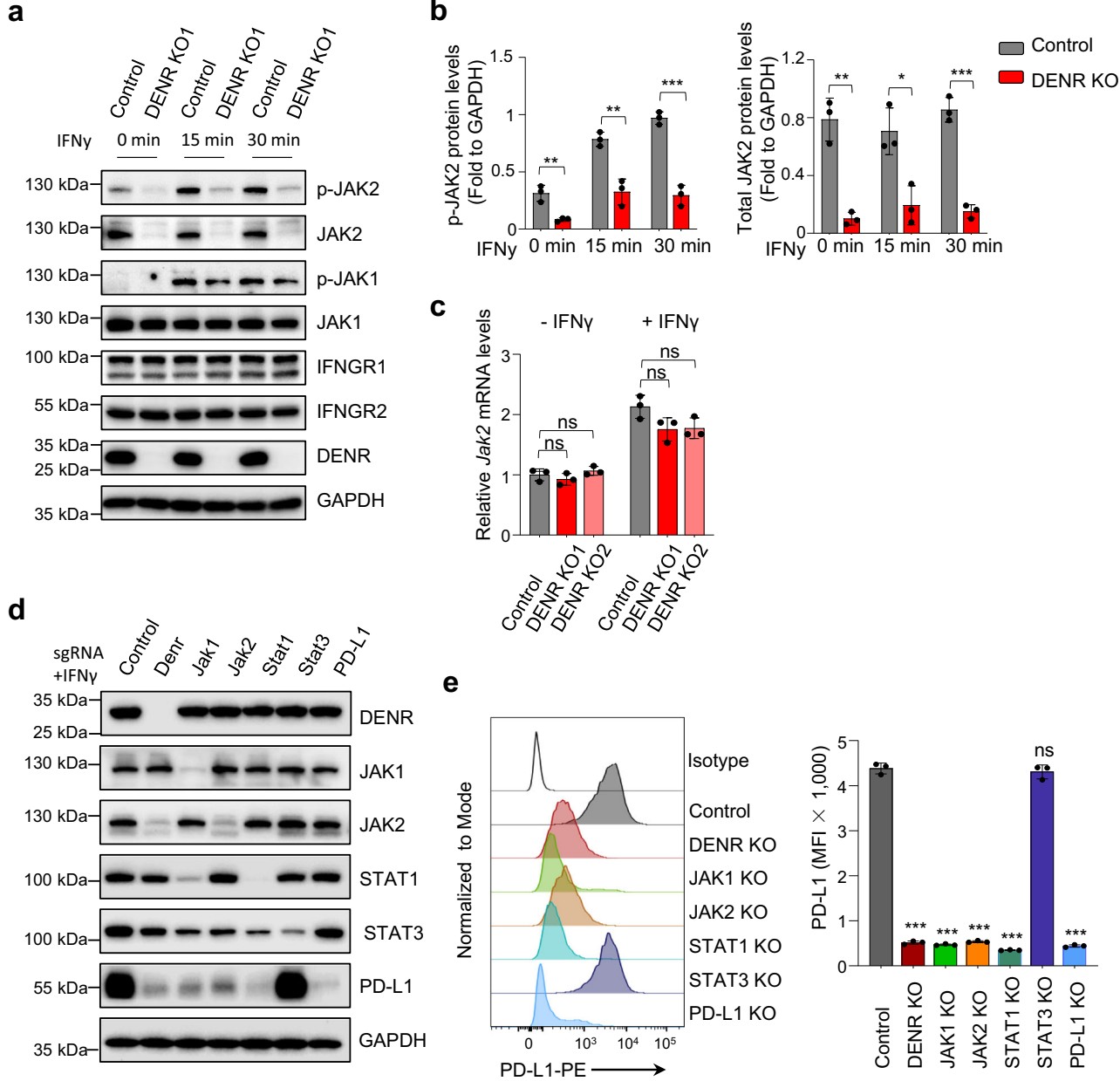

**Fig. 3 JAK2 is translationally controlled by DENR. a** Western blotting analysis of JAK1/p-JAK1, JAK2/p-JAK2, IFNGR1, and IFNGR2 in control or DENR KO MC38 cells, the experiment was repeated independently three times with similar results. **b** p-JAK2 and JAK2 were quantified by normalizing to GAPDH by ImageJ ($n = 3$ biologically independent samples). Data are presented as mean values ± SD, unpaired two-tailed Student's t-test: * $p < 0.05$, ** $p < 0.01$, *** $p < 0.001$. p values from left to right: 0.0053, 0.0035, 0.0004, 0.0016, 0.0136, and 0.0002. **c** RT-qPCR analysis of *Jak2* mRNA levels in control and DENR KO MC38 cells, ± IFNγ (20 ng/ml) for 4 h ($n = 3$ biologically independent samples). Data are presented as mean values ± SD. Two-sided, one-way ANOVA with Dunnett's post hoc test: ns: not significant. *P* values from left to right: 0.5347, 0.6111, 0.0824, and 0.0971. **d** Western blotting validation of DENR, JAK1/2, STAT1/3, and PD-L1 knockout with specific sgRNAs in MC38 cells, + IFNγ (20 ng/ml) for 24 h. The experiment was repeated independently three times with similar results. **e** Surface PD-L1 levels were analyzed by FACS in different KO cell lines as indicated, +IFNγ (20 ng/ml) for 24 h ($n = 3$ independent samples). Results are representative of three biological replicates. Three technical replicates are shown. Data are presented as mean values ± SD. Two-sided, one-way ANOVA with Dunnett's post hoc test: *** $p < 0.001$, ns: not significant. *P* values from left to right: < 0.0001, < 0.0001, < 0.0001, < 0.0001, 0.7065, < 0.0001. Source data are provided as a Source Data file.

culture dishes by Cell Counting Kit 8 (CCK8) assay (Supplementary 5a, b). Then, EGFP-DENR KO and tdTomato-Control WT MC38 cells were evenly mixed and subcutaneously implanted into the flanks of WT C57BL/6 mice, so that the KO and WT tumor cells grew in the same tumor immune microenvironment, eliminating the extrinsic effects. We monitored the tumor volume daily, and selected mice at the early tumor growth time point (day

8) and at the late tumor regression time point (day 12) for sacrifice to analyze the ratio of DNER KO to control tumor cells (Fig. 6a). Interestingly, we found that the percentage of DENR KO tumor cells in all tumors was similar to that of control WT tumor cells on day 8 but much lower than that of WT tumor cells on day 12 (Fig. 6b), indicative of a specific and faster clearance of DENR KO tumor cells in the same tumor microenvironment.

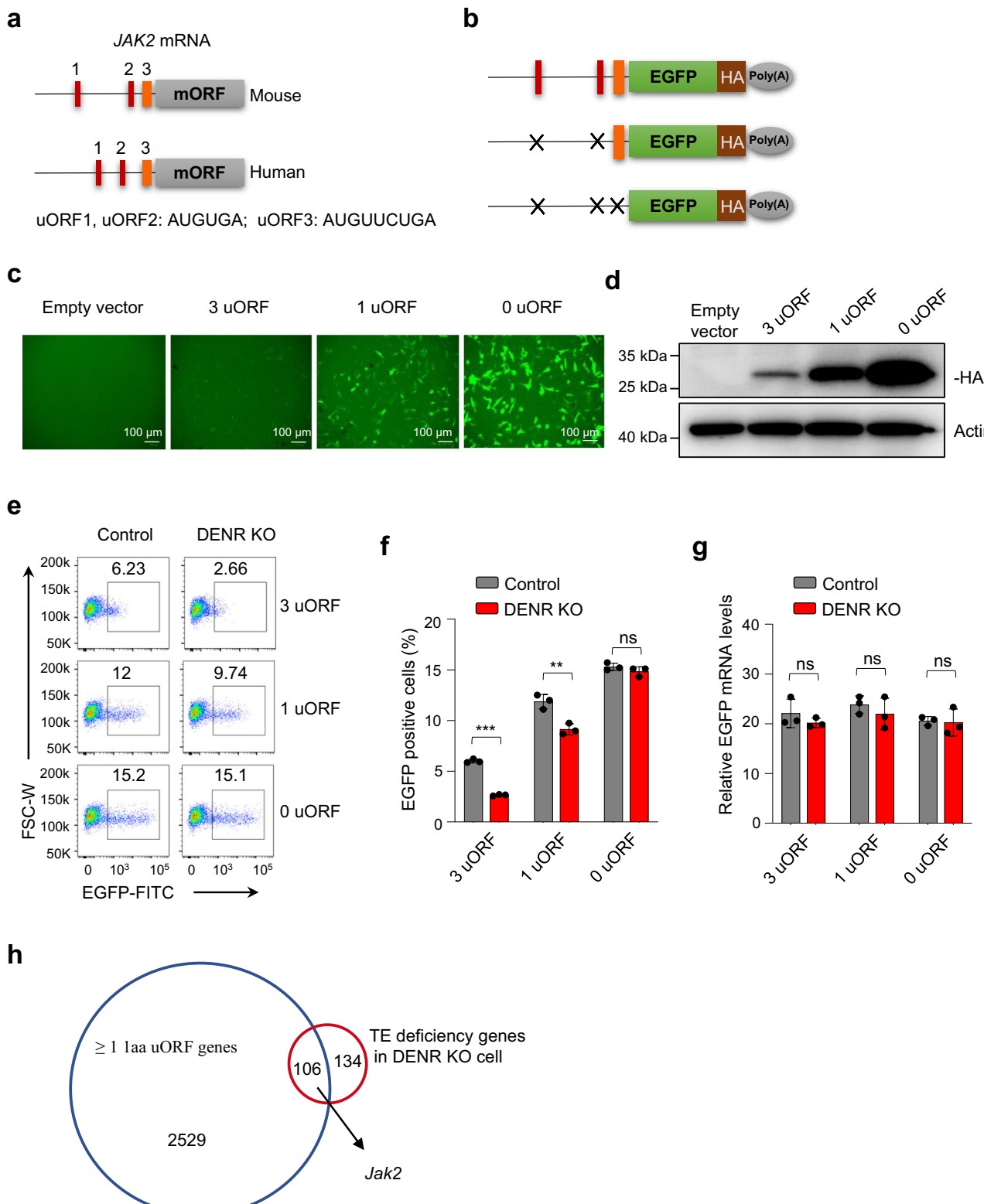

When analyzed by FACS, the PD-L1 expression levels of DENR KO tumor cells were significantly lower than those of control WT cells (Fig. 6c).

As CD8$^+$ T cells are central to tumor killing, and the anti-tumor immunity of CD8$^+$ T cells was inhibited by the expression of PD-L1 in the tumor microenvironment[38], we assess the infiltration and function of the CD8$^+$ T cells in two syngeneic tumor models of both B16/F10 and MC38 tumor cells. We subcutaneously injected DENR KO B16/F10 or MC38 cells into the lower flanks of WT C57BL/6 mice, monitored the daily tumor growth, and sacrificed the mice at the end of the experiment. As expected, the DENR KO tumors grew much slower than WT tumors, and the tumor size of the KO cells was much smaller for B16/F10 tumor (Fig. 6d). More dramatically, MC38 tumors almost did not grow with DENR depletion (Fig. 6e). When B16/F10 tumors were analyzed by FACS, we observed that the

**Fig. 4 DENR regulation of the translation of JAK2 is dependent on three uORFs. a** uORFs of human and mouse *JAK2* 5′ UTR features: 1 amino acid uORFs (red), 2 amino acid uORF (orange). **b** EGFP-HA-SV40 Poly(A) reporters of murine WT *Jak2* 5′UTR with three uORFs or mutated 5′UTR with 1 uORF or 0 uORF. **c** WT B16/F10 cells (1 × 10[7]) were divided into six-well plates; 24 h later, 650 ng of plasmid with individual reporters with 1 μl Lip3000 was added, and the cells incubated for 14–16 h. The EGFP was analyzed by fluorescence microscope. Scale bars, 100 μm. The experiment was repeated independently three times with similar results. **d** The protein levels of HA-tag as performed in **c** were analyzed by western blotting. The experiment was repeated independently three times with similar results. **e**, **f** Control or DENR KO B16/F10 cells were incubated with reports as described, and the percentage of EGFP positive cells were analyzed by FACS (*n* = 3 independent samples). Results are representative of three biological replicates. Three technical replicates are shown. Data are presented as mean values ± SD, unpaired two-tailed Student's t-test: ** *p* < 0.01, *** *p* < 0.001, ns: not significant. *p* values from left to right: < 0.001, 0.0061, 0.2399. **g** The corresponding mRNA levels of EGFP reports in **e** were measured (*n* = 3 independent samples). Results are representative of three biological replicates. Three technical replicates are shown. Data are presented as mean values ± SD, unpaired two-tailed Student's t-test: ns, not significant. *p* values from left to right: 0.8567, 0.4271, and 0.3341. **h** Venn diagram shows the overlap of more than 1 aa (amino acid) uORF genes with the TE (translation efficiency) impaired genes in DENR KO NIH3T3 cells. Source data are provided as a Source Data file.

percentage of CD4+ T cells among the immune cell population was similar in the KO and control tumors, the percentage of CD8+ T cells was significantly higher in DENR KO tumors than WT tumors (Fig. 6f, g). Next, to prove that the more efficient killing of DENR KO tumors over WT tumors was indeed due to increased tumor infiltration and enhanced killing of CD8+ T cells, we subcutaneously implanted DENR KO and control cells into the flanks of RAG1−/− mice in which the T cells and B cells were depleted. We observed that the growth of DENR KO and WT tumors in the RAG1−/− mice no longer showed significant differences without the existence of T cells for B16/F10 tumors, and slight difference for MC38 tumors (Supplementary Fig. 5c, d).

It is established that PD-L1 in tumors inhibits tumor immune clearance[38], and our results showed that DENR KO dramatically attenuated tumor growth with decreased PD-L1 expression and increased CD8+ T cells infiltration. Together, our data indicate that DENR expression in tumors supports tumor evasion by regulating PD-L1 expression, and DENR KO assists faster tumor immune ejection, suggesting that DENR is a promising immunotherapy target.

**DENR correlates with patient survival in immunotherapy**. As tumor cells rely on the expression of PD-L1 to resist attack by CD8+ T cells[38], we assess the direct tumor-killing ability of CD8+ T cells by co-cultured with DENR KO or control MC38 cancer cells. The results showed that in OVA expressing cancer cells, DENR depletion resulted in weaker p-STAT1 signaling, lower PD-L1 expression, and significant higher percentage of apoptotic cells (Fig. 7a–c and Supplementary Fig. 5e–g). Notably, adding anti-PD-1 to the co-culture to block PD-L1 also increased the apoptotic population of control cells (Fig. 7a, b). Together, these results suggest that DENR depletion in tumors reduces the expression of PD-L1 upon CD8+ T cell treatment and, thus, enhances CD8+ T cell anti-tumor activities.

In clinical studies, anti-PD1/PD-L1 therapy has benefited patients with diverse types of cancers. However, the low response rates to PD1/PD-L1 therapy in some tumor types and patients highlight the urgent clinical need to identify other immunotherapy targets or combination strategies[7]. To evaluate whether DENR can serve as a potential target or marker in anti-PD1/PD-L1 therapy, we analyzed three independent clinical studies of melanoma patients under anti-PD-1[39,40], or anti-CTLA-4[41] treatment regimens (Fig. 7d–f). We found that lower DENR expression in tumors correlated with better survival outcomes of patients given anti-PD-1 therapies (Fig. 7d, e), but not with anti-CTLA-4 therapy (Fig. 7f), demonstrating the potential benefits of using DENR as an immunotherapy target or marker.

## Discussion

RBPs play an essential role in RNA metabolism, and aberrant RNA-RBP networks are related to cellular dysfunction and

human diseases[42–46]. However, which RBPs affect tumor immune evasion by regulating PD-L1 is largely unknown. Here, we described identifying RBP-DENR as a regulator of PD-L1 by employing pooled RBP CRISPR-Cas9 screening. Mechanistically, we found three uORFs in the 5′ UTR of *JAK2*, and loss of DENR stalled the 40S ribosomes on these uORFs and reduced the translation of JAK2, which in turn impaired the JAK2-STAT1 signaling pathway and repressed PD-L1 expression in vitro and in vivo (Fig. 7g).

JAK2 is involved in a subset of cytokine receptor signaling pathways, including those involved in cell growth, development, and differentiation, such as growth hormone; erythropoietin; type-II receptors, IFNα, IFNβ, and IFNγ; and a wide range of interleukins[47]. JAK2V617F is a highly recurrent mutation in myeloproliferative diseases, in which the normal valine on the 617th amino acid in the coding region is replaced by phenylalanine[48]. This change activates JAK2 and improves PD-L1 expression, mediating abnormal proliferation and immune escape in myeloproliferative neoplasms[48]. We observed that DENR depletion almost abolished the translation of JAK2, suggesting DENR could be a potential target in JAK2V617F-mediated diseases. Furthermore, as the three uORFs found in *JAK2* constitutively repressed the translation of JAK2, and with clinical sequence research reporting an uORF mutation of *JAK2* in a patient with chronic lymphocytic leukemia[49], it appears the mutation of uORFs in *JAK2* is a high-risk factor, and more in-depth sequencing of the 5′ UTR is needed in patients with myeloproliferative diseases.

In our study, we have identified that DENR positively regulates PD-L1 expression through IFNγ-JAK2-STAT1 signaling, DENR high patients are less sensitive to PD-L1/PDL1 immune check blockade (ICB) therapy. The results seem inconsistent to some previous study showing some PD-L1 high patients were more sensitive to PD-L1/PDL1 blockade. However, PD-L1/PDL1 blockade therapy only benefit ~20% cancer patients, and among the remaining patients, many of them have mid-high PD-L1 expression[7]. In Tumor Microenvironment, a major source of IFNγ are T cells. Infiltrating T cells secrete more IFNγ, which stimulate tumor cells via JAK-STAT signaling to overexpress PD-L1, as well as many other known and unknown molecules, to inhibit T cell function[50]. This could explain that only ~ 20% patients could benefit from ICB. It has been well studied that interfering with tumor IFN-JAK-STAT signaling combined with PD-1/PD-L1 immune check blockade (ICB) monotherapy was more effective than only ICB monotherapy[50]. Our current study proves that DENR directly regulates JAK2, and downregulation of DENR will inhibit IFN-JAK-STAT signaling and its downstream PD-L1 and many other IFN response genes. Thus, DENR low patients with weakened JAK-STAT signaling could be more sensitive to ICB therapy. Multiple previous studies also have reported that candidate genes positively regulate tumor expression of PD-L1 is associated with PD-1 blockade resistance[51,52]. However, the clinical data is limited in our study, and the DENR

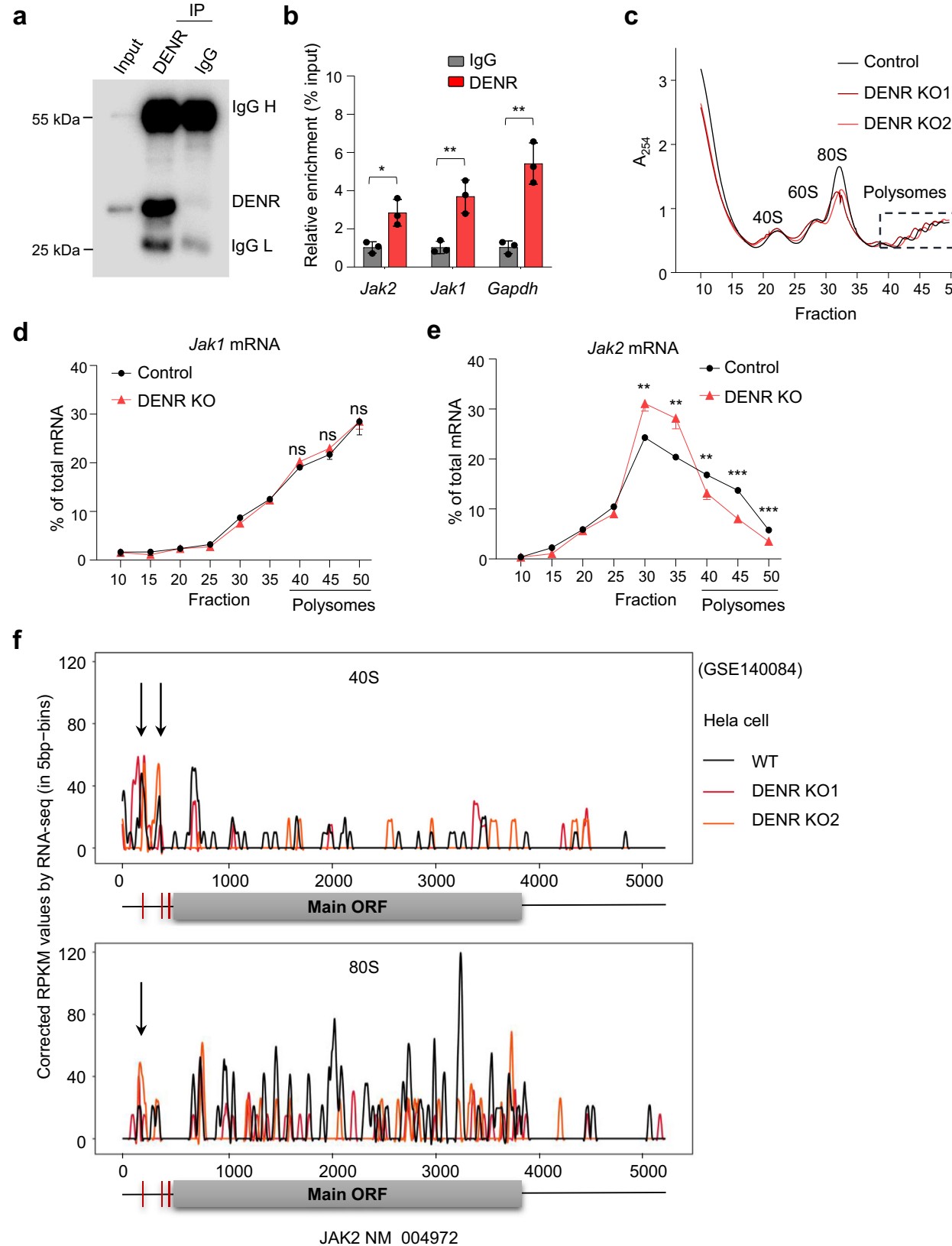

JAK2 NM_004972

function in tumor microenvironment of the tumor patients is very complex and is in need of further investigation.

DENR associates ribosomes but does not have the specific RNA binding motif that is seen in other RBPs[53,54]. However, the function of DENR is also highly correlated with particular uORF RNA sequences[24]. Consistent with our findings, some studies

reported that DENR shows the highest regulation efficiency at start-stop uORFs and weaker efficiency at longer uORFs[24], and the Kazok sequence before uORFs and penultimate codons are also involved[24,30]. This suggests the mechanisms of translation re-initiation of ribosomes to overcome uORFs are complex and may involve multiple factors; therefore, further study is needed.

**Fig. 5 DENR depletion stalls ribosomes on uORFs. a** DENR was immunoprecipitated with the anti-DENR antibody, the experiment was repeated independently three times with similar results. **b** RIP analysis of DENR association with mRNAs in MC38 cells. DENR-bound mRNAs were detected by RT-qPCR ($n = 3$ biologically independent samples). Data are presented as mean values ± SD, unpaired two-tailed Student's t-test: * $p < 0.05$, ** $p < 0.01$. $p$ values from left to right: 0.0144, 0.0074, and 0.0026. **c** Representative traces of polyribosome profiles obtained from control or DENR KO MC38 cells. The experiment was repeated independently three times with similar results. **d** RT-qPCR analysis of *Jak1* mRNA levels in each fraction obtained from control or DENR KO MC38 cells ($n = 3$ independent samples). Results are representative of two biological replicates. Three technical replicates are shown. Data are presented as mean values ± SD, unpaired two-tailed Student's t-test: ns, not significant. $p$ values from left to right: 0.1344, 0.1233, and 0.9869. **e** RT-qPCR analysis of *Jak2* mRNA levels in each fraction obtained from control or DENR KO MC38 cells ($n = 3$ independent samples). Results are representative of two biological replicates. Three technical replicates are shown. Data are presented as mean values ± SD, unpaired two-tailed Student's t-test: ** $p < 0.01$, *** $p < 0.001$. $p$ values from left to right: 0.0015, 0.0032, 0.0086, <0.0001, 0.0006. **f** 40S and 80S ribosome occupancy of *JAK2* mRNA transcripts in DENR KO versus WT Hela cells. Source data are provided as a Source Data file.

DENR is highly expressed in multiple tumor types and indicates the poor prognosis of patients[55]. In this study, we uncovered the function of DENR in regulating PD-L1 expression in immune evasion, and the results indicate that DENR is a potential therapeutic target, either in isolation or in combination with current PD-L1/PD-1 blockade therapies.

## Methods

**Mice**. All the wild-type mice at 6- to 8-week-old of age were maintained under specific pathogen-free conditions, and all animal procedures were performed in ethical compliance and with approval by the Institutional Animal Care and Use Committee (IACUC) of Shanghai Jiao Tong University School of Medicine.

**Cell culture**. HEK293T, RAW264.7, MC38, B16/F10, HT29, and A375 cells were obtained from ATCC. Cells were cultured in RPMI-1640 or DMEM supplemented with 100 IU/ml Penicillin, 100 μg/ml Streptomycin and 10% fetal calf serum.

**Cloning**. sgRNA oligos were synthesized and cloned into the LentiGuide-puro (#52963, Addgene) or LentiCRISPR v2-Blast (#52963, Addgene) via the two BsmBI sites. The EGFP-sgRNA plasmid in this study was a gift from Jiyu Tong, and we replaced the EGFP with a tdTomato to generate a tdTomato-sgRNA plasmid. The WT *Jak2* 5′ UTR (NM_008413.4) and mutations were synthesized and cloned into pCMV-EGFP-HA-C (P0916, MiaoLing Plasmid Sharing Platform) by EasyGeno Assembly Cloning Kit (VI201, Tiangen) according to the manufacturer's recommendations.

**Lentiviral production and transduction**. Lentivirus was produced by transfection of HEK293T cells with a transfer vector of sgRNA expression plasmids or lentiCas9-Blast (#52962, Addgene), and the packaging plasmids psPAX2 (#12260, Addgene) and pMD.G (#12259, Addgene) at a 4:3:1 ratio. Transfection was performed using Roche reagent as recommended by the manufacturer. The viral supernatant was collected at 48 h and 72 h following transfection, filtered through a 0.45 μm filter, and added to target cells.

**CRISPR screen**. A 10 sgRNA-per-gene CRISPR/Cas9 deletion library was designed to target 1467 RBP genes using CRISPR-FOCUS (cistrome.org/crispr-focus/), the library contained negative control sgRNAs: no binding sites in the genome, and positive control sgRNAs in IFNγ pathway (Supplementary Data 1). RAW264.7 cells were transduced with a lentiviral vector encoding Cas9 and selected with blasticidin (5 μg/ml) for 7 days. $5 \times 10^7$ Cas9-expression RAW264.7 cells were infected with the pooled lentiviral RBP sgRNA library at a multiplicity of infection of 0.3 and selected with 5 μg/ml puromycin for 72 h, commencing 48 h after transduction. Rare PD-L1 low and PD-L1 high cells were enriched by FACS sorting at day 7. For the sorts, cells were pre-treated with 20 ng/ml IFNγ for 4 h, stained with PE-conjugated anti-PD-L1 antibody for 20 min, and washed with PBS prior to sorting for PD-L1 low and high cells. Genomic DNA was extracted from both the sorted cells and the unsorted cells. sgRNA sequences were amplified by two rounds of PCR, and purified for sequencing on Illumina Hiseq.

**Immunoblotting and immunoprecipitation**. For western blotting analysis, cells were lysed in RIPA buffer (P0013B, Beyotime) containing protease and phosphatase Inhibitor cocktail, EDTA free (78441, Thermo Fisher Scientific) for 30 min on ice followed by pelleting of insoluble material by centrifugation. Protein concentration was determined using a BCA assay. For co-immunoprecipitation experiments, 10 million cells were lysed in 1 ml NP40 buffer (P0013F, Beyotime) containing protease and phosphatase Inhibitor cocktail, EDTA free, for 30 min on ice. After concentration, supernatants were incubated with Rabbit anti-DENR, Rabbit anti-JAK2, or Rabbit normal IgG for 4 h at 4 °C, followed by addition of 25 μl protein A/G beads (B23201, Bimake) and incubation for a further 2 h. After

five washes in NP40 buffer, beads were eluted in 100 μl IgG elution buffer, PH2.0 (21028, Thermo Fisher Scientific) and neutralized with 15 μl 50 mM NaOH.

**Crosslinking and RNA-IP**. MC38 cells were grown to confluency in 10 cm dish, cells were washed twice with cold PBS and irradiate uncovered with 0.2 Jcm$^{-2}$ total energy of 245 nm UV light. Cells in one dish were then scraped in 1 ml RNA-IP lysis buffer (25 mM Tris-HCl (pH 7.5), 1% NP-40, 150 mM NaCl, 1.5 mM MgCl2, 1 mM DTT, Protease and Phosphatase Inhibitor Cocktail, EDTA-free and 100 U/ml RNase inhibitor (Vazyme)). Lysates were sonicated (30 s on/30 s off) for 3 cycles (Bioruptor® Pico sonication device) and cleared by centrifugation and split into input (100ul) and IP samples (800ul). The IP samples were incubated with 2 μg anti-DENR or Rabbit normal IgG for 4 h at 4 °C. Immunocomplexes were precipitated by incubation with protein A/G beads for 2 h at 4 °C. Beads were washed three times with low salt wash buffer (25 mM Tris-HCl (pH 7.5), 0.05% NP-40, 150 mM NaCl, 1.5 mM MgCl2 and 100 U/ml RNase inhibitor), followed by 3 washes with high salt wash buffer (25 mM Tris-HCl (pH 7.5), 0.05% NP-40, 300 mM NaCl, 1.5 mM MgCl2 and 100 U/ml RNase inhibitor). RNA in input or IP samples was eluted by Trizol reagent according to the manufacturer's recommendations. 2 μl RNA were subjected to reverse transcription using TransScript All-in-One First-Strand cDNA Synthesis SuperMix (AT341-03, Transgen) according to the manufacturer's instructions, followed by qPCR analysis.

**In vitro killing assay**. MC38 cells with or without OVA expression were infected with Cas9 and sgRNAs expression lentivirus to generate stable MC38 ± OVA Control or DENR KO cells. The spleen of OT-1 mice was dissociated and cultured with 5 μg/ml OVA $_{(257-264)}$ peptides and 4 ng/ml IL-2 for three days, and changed to fresh media contain 4 ng/ml IL-2 for another three days. After amplification, CD8$^+$ T cells were purified by negative selection with magnetic beads (8804-6825-74, ThermoFisher). After purification, $6 \times 10^5$ CD8$^+$ T cells were coculture with $2 \times 10^5$ MC38-OVA Control or DENR KO cells (Effector: Target ratio: 3:1) in 24 wells plates, anti-PD1 (10 μg/ml) was added in MC38-OVA Control cells as a positive control. The cells were harvested after 16 h and labeled with 7-AAD, Caspase-3/7 or PD-L1 to detect by FACS, the gating strategy was shown as in Supplementary Fig. 4b.

**Flow cytometry**. Cells from the plates or tumors were isolated and resuspended in PBS plus 1% FBS and 5 mM EDTA. After staining with the fluorescently conjugated antibodies, flow cytometry data were acquired on BD LSRFortessa X-20 or Accuri C6 Plus and analyzed using FlowJo software (Tree Star), the gating strategy in analysis tumor tissues were shown as in Supplementary Fig. 4a.

**Tumor models**. Female WT C57BL/6 mice (6–8 weeks) and female RAG1$^{-/-}$ mice (6–8 weeks) were purchased from Shanghai Model Organisms Center, the animals were maintained in specific-pathogen-free facilities. MC38 or B16/F10 cells were washed and resuspended in PBS at $5 \times 10^6$/ml, mice were injected with 100 μl of total $5 \times 10^5$ cells in the flank subcutaneously of WT C57BL/6 mice. Tumors were measured every 1–2 days (length × width) with a digital caliper. RAG1$^{-/-}$ mice were given $1 \times 10^5$ control or DENR KO cells, and tumors were measured every day. Tumor volume was determined using the formula: $0.52 \times L \times W^2$. Mice were sacrificed before tumors ulceration, and in the study, all the tumor diameters did not exceed 20 mm that was the maximal tumor diameter permitted by the IACUC of Shanghai Jiao Tong University School of Medicine. For mixture EGFP-DENR KO and tdTomato-Control cells, $5 \times 10^5$ EGFP and $5 \times 10^5$ tdTomato cells were mixed and injected in the flank subcutaneously, tumors were measured every day, volumes were determined as described.

**Antibodies**. The antibodies used in the study were listed in the Supplementary Table 1.

**sgRNA and RT-qPCR sequence**. The oligonucleotides for sgRNA cloning and PCR primers used in the study were listed in the Supplementary Tables 2 and 3.

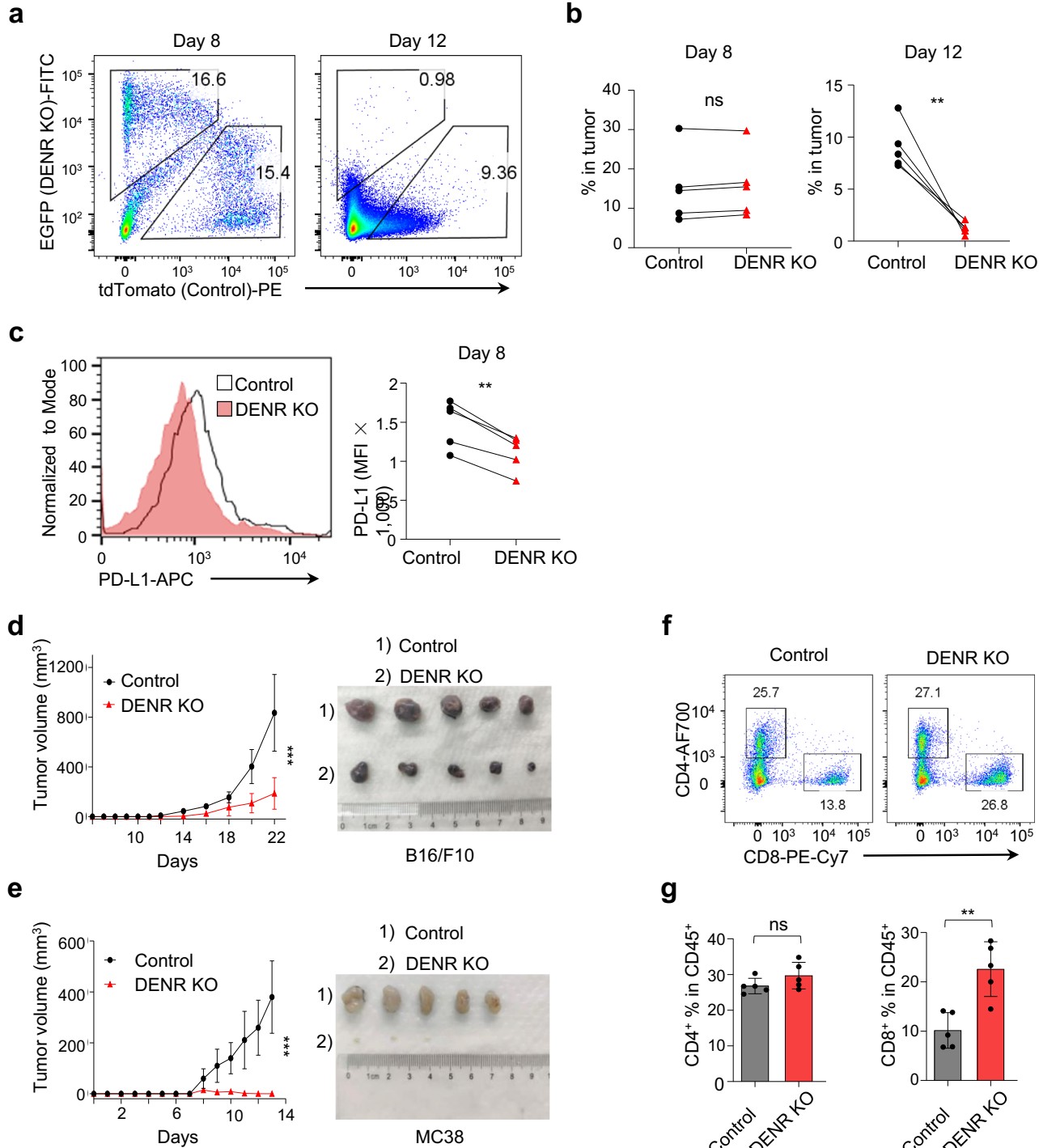

**Fig. 6 DENR depletion reduces tumor growth and PD-L1 expression in vivo. a, b** Control and DENR KO MC38 cells stably expressing tdTomato or EGFP were mixed at a 1:1 ratio and administered subcutaneously into 6–8 weeks old female WT C57BL/6 mice ($n = 5$ mice). The composition of EGFP and tdTomato-positive cells in tumors was analyzed on day 8 or 12 by FACS. Data are presented as single values and are representative of three independent experiments. Paired two-tailed Student's t-test: ns, no significant, ** $p < 0.01$. $p$ values from left to right: 0.1078, 0.0032. **c** Cell surface PD-L1 on control or DENR KO MC38 cells in tumors on day 8 as shown in **a** was analyzed by FACS ($n = 5$ tumors). Data are presented as single values and are representative of three independent experiments with similar results. Paired two-tailed Student's t-test: ** $p < 0.01$. $p = 0.0017$. **d, e** Female WT C57BL/6 mice (6–8 weeks) were given $5 \times 10^5$ control or DENR KO tumor cells, and tumors were measured every 1–2 days ($n = 5$ mice). Tumor growth is representative of three independent experiments with similar results. Data are presented as mean values ± SD, two-way ANOVA test: *** $p < 0.001$. $p$ values from top to bottom: < 0.0001, <0.0001. **f, g** FACS analysis of intratumor CD4$^+$ and CD8$^+$ T cells in B16/F10 tumors ($n = 5$ tumors) as shown in (**d**). Data are presented as mean values ± SD and are representative of three independent experiments with similar results. Unpaired two-tailed Student's t-test: ns, no significant, ** $p < 0.01$. $p$ values from left to right: 0.1712, 0.0030. Source data are provided as a Source Data file.

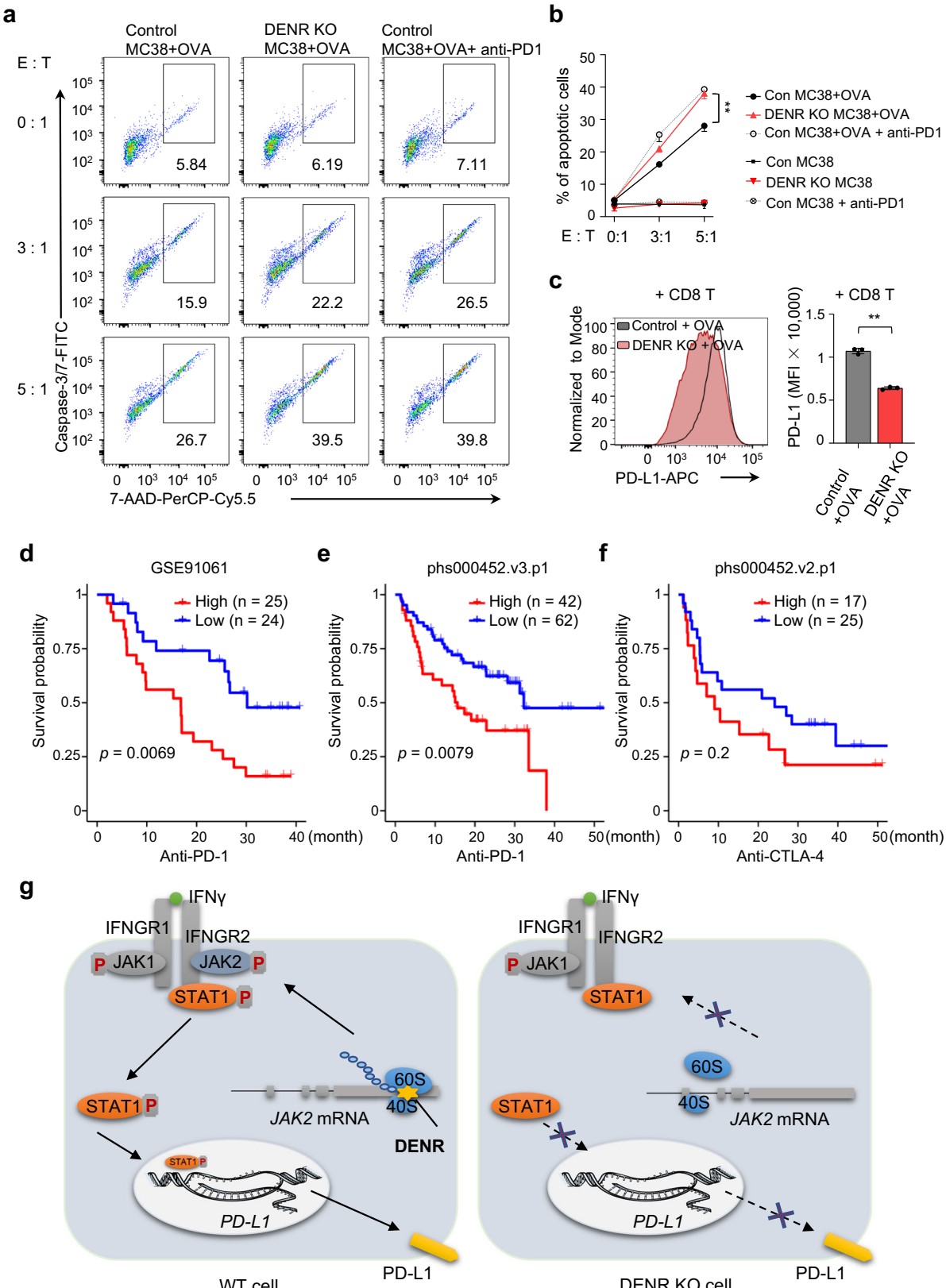

**RNA extraction for RNA-seq and quantitative real-time PCR.** For RNA-seq, MC38 control and DENR KO cells were stimulated with IFNγ (20 ng/ml) for 4 h and total RNA was extracted using TRIZOL (Invitrogen #15596026). RNA purification, reverse transcription, library construction, and sequencing were performed by Novogene Co., Ltd. (Shanghai, China) according to the manufacturer's instructions (Illumina). Briefly, mRNA was purified from total RNAs by using poly-T oligo-attached magnetic beads and fragmented by the fragmentation buffer.

First strand cDNA was synthesized using random hexamer primer and M-MuLV Reverse Transcriptase, then using RNase H to degrade the RNAs. Second strand cDNA synthesis was subsequently performed using DNA Polymerase I and dNTP. Remaining overhangs were converted into blunt ends via exonuclease/polymerase activities. After adenylation of 3′ ends of DNA fragments, adaptor with hairpin loop structure were ligated to prepare for hybridization. In order to select cDNA fragments of preferentially 370–420 bp in length, the library fragments were

**Fig. 7 DENR depletion enhances the anti-tumor killing ability of CD8+ T cells. a, b** Control or DENR KO MC38 cells with or without expression OVA were co-cultured with OT-1 CD8+ T cells at an effector (E) to target (T) ratio of 0:1, 3:1, or 5:1 for 16 h; anti-PD-1 (10 μg/ml) was added as a positive control. The apoptosis cells were measured by flow cytometry ($n = 3$ independent samples). Results are representative of three biological replicates. Three technical replicates are shown. Data are presented as mean values ± SD, unpaired two-tailed Student's t-test at E: T = 5:1: ** $p < 0.01$. $p = 0.0020$. **c** Control or DENR KO MC38 cells with expression OVA were incubated with OT-1 CD8+ T cells at an effector to target ratio of 3:1 for 16 h and surface PD-L1 were measured by flow cytometry ($n = 3$ independent samples). Results are representative of three biological replicates. Three technical replicates are shown. Data are presented as mean values ± SD, unpaired two-tailed Student's t-test: *** $p < 0.001$. $p < 0.0001$. Kaplan–Meier curves show overall survival in the DENR-high (red) and -low (blue) subgroups after anti-PD-1 blockade immunotherapy in **d, e** or anti-CTLA-4 blockade immunotherapy in (**f**). $P$ value is calculated using the log-rank test. **g** DENR binds to 40S ribosomes to bypass three uORFs in *JAK2* mRNA. In the absence of DENR, JAK2 was not effectively translated and with reduced PD-L1 transcription by impairing the IFNγ signaling conduction. Source data are provided as a Source Data file.

purified with AMPure XP system (Beckman Coulter, Beverly, USA). Then PCR amplification, the PCR product was purified by AMPure XP beads, and the library was finally obtained. The library was initially quantified by Qubit2.0 Fluorometer, then diluted to 1.5 ng/μl, and the insert size of the library is detected by Agilent 2100 bioanalyzer. After the library is qualified, the different libraries are pooled according to the effective concentration and the target amount of data off the machine, then being sequenced by the Illumina NovaSeq 6000. The end reading of 150 bp pairing is generated.

For quantitative real time PCR, total cellular RNA or RNA in polysome fraction was extracted using TRIZOL (Invitrogen #15596026). mRNA was reversely transcribed into cDNA using HiScript III 1st Strand cDNA Synthesis Kit (Vazyme, R312-01). Fluorescence real-time PCR was performed with the ChamQ Universal SYBR qPCR Master Mix (Vazyme, q711-02) with the CFX384 Touch Real-Time PCR Detection System (Bio-Rad, USA). β-actin was used as an internal control for normalization.

**Statistical analysis.** Survminer_0.4.9 package was used to determine the cutoff point of survival information for each dataset based on the association between DENR expression and patient overall survival. To find the maximum rank statistic and reduce the calculated batch effect, the "surv-cutpoint" function was used to dichotomy DENR expression, and all potential cutting points were repeatedly tested, then the patient samples were divided into the high-DENR group and the low-DENR group according to the maximum selected log-rank statistics. Kaplan–Meier comparative survival analyses for prognostic analysis were generated, and the log-rank test was used to determine the significance of the differences. For GSEA analysis the gene set "c2.all.v7.5" was downloaded from the MSigDB database. All statistical analysis was two-side and considered $P < 0.05$ as statistical significance.

**Reporting summary.** Further information on research design is available in the Nature Research Reporting Summary linked to this article.

## Data availability
The RNA-seq data generated in this study have been deposited in the GEO database under accession code GSE183980. The CRISPR screen data generated in this study have been deposited in the GEO database under accession GSE184048. For GSEA analysis the gene set "c2.all.v7.5" was downloaded from the MSigDB database [http://www.gsea-msigdb.org/gsea/index.jsp]. The remaining data are available within the Article, Supplementary Information or Source Data file. Source data are provided with this paper.

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

## Acknowledgements

We thank Kaiqiong Mao, Mei Yang, Jing Zhou, Haixin Li, and all other members of the Hua-Bing Li lab for discussions. This work was supported by the National Natural Science Foundation of China (82030042/32070917 to H.-B.L., 81725004 to H.L.), Shanghai Science and Technology Committee (grant no. 20JC1417400/201409005500/20JC1410100 to H.-B.L.; 20MC920200 and19XD4010000 to H.L.).

## Author contributions

H.-B.L. conceived the project and designed the research. B.C., X.H., and Y.Z. designed and performed the experiments. B.C., J.H., X.H., H.C., R.B., and Y.Y. analyzed and interpreted the data. H.C. and R.B. helped with analyzing RNA-seq and ribosome profiling data. H.-B.L., H.L., W.C., Y.Y., and M.Z. discussed the projects. H.-B.L. and B.C. wrote the manuscript. This study was supervised by H.L. and H.-B.L. All authors read and approved the final manuscript.

## Competing interests

The authors declare no competing interests.
