## [Peer Review File · Nature Communications]

Reviewers' Comments:

Reviewer #1:

Remarks to the Author:

The paper is addressing an interesting question in determining the patients that could benefit from anti-PD-1/PD-L1 blockad and unravelling the molecular mechanism would be important. However, there are many inconsistencies in the work and the authors do not really address the question of why patients with high DENR are less sensitive to PD-L1 blockade. In addition, the paper is rather rambling and would benefit from some of the figures, which essentially show negative data, being moved to the supplementary section to provide a more focussed story.

Major comments

1. Regarding the statistical analysis in Figures 1G, 1I, 2B, 3H, 3G and 7B - multiple comparisons (multiple t-tests) have been made between the data in each figure panel. Running multiple t-tests in this manner increases the chance of error, therefore, statistical analysis should take these multiple comparisons into account by using ANOVA with Dunnett's post hoc test.

2. In the introduction the authors explain that IFN gamma is a known activator of the JAK-STAT pathway, yet Fig 3a shows that IFN gamma does not phosphorylate JAK2 in control sgRNA cells (and Fig 3b shows no phosphorylation of JAK1 with IFN gamma). Can the authors explain why DENR KO cells behave this way? Additionally, densitometry (for phospho proteins normalised to the total level of the protein) need to be carried out and statistical analysis of the western blot bands is required as some changes are very subtle.

3. Figure 4. The appears to be almost no variation on the bar charts shown for figure 4G which surprising. Are these technical repeats or biological repeats?

4 Related to Figure 5c: the author's state "Together, these data exclude the possibility that DENR has a broad effect on the translation of all mRNAs...". However, polysome profiling does not clearly show changes in translation elongation. For example, stalled (non-elongating) ribosomes would remain bound to RNA and appear as heavy polysomes, which would be interpreted as active translation. Moreover, there is a decrease in 80S in the DENR KO cells, but no corresponding change in 40S or 60S or heavy polysomes. This raises a number of questions, such as where are the free 80S ribosomes going? Are they being degraded or are they being recruited to insoluble granules? if either of these were a possibility then translation rates would most likely be decreased.

Therefore, to validate the statement and exclude that DENR has a broad effect on translation of all mRNA the authors should also:

- Investigate the rate of translation initiation (i.e. with radiolabelled methionine incorporation)
- Determine the phosphorylation status of translation factors that are known to inhibit protein synthesis (eIF2a, 4EBP1, eEF2) in the control and KO cell lines.
- Although ribosome profiling of other studies has been analysed it is important to carry out RT-qPCR on polysome fractions for individual mRNAs of interest (such as Jak1/2) to determine if they are associated with heavy polysomes (i.e. being actively translated) in this cell line KO model.

5. There are inconsistencies in the way PD-L1 levels are shown. Some figures use only total PD-L1 in wester blots (i.e. Fig2A), whereas others quantify surface PD-L1 (i.e. Fig 1). Why is cell surface quantification used in some experiments and not others? Is there a population of PD-L1 in the cell that is not presented on the cell surface that would be detected in the western blot? It is important to include both types of analysis throughout the manuscript.

6. Jak1 mRNA is also bound by DENR, yet DENR KO does not have any impact on JAK1 protein or mRNA levels. Could the authors expand on why this may be the case?

Minor comments

- Extended figure 2 – no key for colours of bars.
- Figure 4F and 4G – the labelling of uORFs for 4F is actually part of panel 4g and so is unclear.

- Figure 6J and 6K are referenced in the text before 6I.
- Please add a label to distinguish the studies illustrated in figure 7E and 7F.
- Figure 7H is not referenced in the main text.

Reviewer #2:

Remarks to the Author:

This manuscript provides evidence supporting the notion that an RNA-binding protein (RBP), density regulated re-initiation and release factor (DENR), regulates PD-L1 expression by controlling the translation of JAK2, a component of the upstream signaling pathway known to regulate PD-L1. The authors identified DENR in a targeted CRISPR/Cas9 screen for RBPs that may influence PD-L1 expression, using a pool of sgRNAs targeting 1467 RBPs. They found that DENR depletion reduced PD-L1 expression in stable Cas9-expressing RAW264.7 cells after IFN γ treatment. They further determined that JAK2 protein translation is specifically reduced in DENR depleted cells, and that 3 uORFs present in the 5' region of JAK2 mRNA mediate the effect.

A major concern is that DENR as a general RBP should have broad targets and is unlikely a specific regulator of JAK2, much less PD-L1. Depleting DENR may have many unintended off-targets, thus reducing its potential as a therapeutic target for immunotherapy.

Other concerns are as the following.

Methods: Need to provide the design and sequence information about the sgRNAs used in the screen and specific knock down experiments.

IFN γ treatment: "20 ng/ml treatment for 4 h" in Methods for screen, but "for 24 h" in Extended Fig. 2A. In other places, no specifics were mentioned. Need to provide details and rationale. Results for 4 h and 24 h treatment should be very different on JAK/STAT signaling.

Fig. 1. DENR knockout (KO): How were the KO cells created? Were they stable lines or pooled polyclonal cells? If stable lines, need to provide sequence information regarding genomic deletions. Should provide sequence information for *denr* sgRNA1-3.

Do JAK and STAT KO affect PD-L1 expression? How do they compare with *denr* KO?

Fig. 2. Not clear what was done in "partly rescued by the transient overexpression of an sgRNA against DENR (Extended Data Fig. 2b), which excluded the off-targeting of JAK2 by DENR sgRNAs." Different sgRNAs or JAK2 overexpression? What exactly were overexpressed?

JAK2 IP: Did the IP pull down any JAK2 interactors? How do we know the IP worked?

Fig.4. Should have systematic scan for the presence of uORFs in genes that are affected by DENR knock down vs. those not affected. It is unlikely that JAK2 is the only target regulated by DENR.

Fig. 5. Need to explain/analyze why DENR associated with GAPDH without affecting its expression. Any uORFs present in *Gapdh* transcript?

Reviewer #3:

Remarks to the Author:

In this study, the authors have reported that an RNA binding protein, DENR, attenuates IFN γ -induced expression of PD-L1 by inhibiting translation of JAK2 mRNA. They have shown that DENR directly binds to the uORFs in the 5'UTR of JAK2 mRNA, and thereby helps the 40S ribosomes to bypass the uORFs and initiate translation of JAK2. Consequently, the upregulation of JAK2 mediates IFN γ -induced expression of PD-L1 to evade the killing of host anti-tumor CD8+ T cells. These findings are well-designed and precise. However, there are some minor issues that should be addressed before its acceptance by Nat Commun.

1. As DENR binds to uORFs to help mRNA translation, it is critical to examine whether knockout of DENR has any effects on tumor cell growth or death in cultures and in immune-compromised mice.
2. In Fig 7C-E, the authors detected the apoptosis rate to assess the killing of tumor cells by CD8 + T cells. The difference is rather minor. CD8+ T cells-mediated tumor killing also induces necroptosis, ferroptosis and pyroptosis. It would be more helpful if the authors use a board cell-death measurement to assess the ratio of cell death.
3. DENR binds to mRNAs of different genes (for example, JAK2, ATF4 and GAPDH in this study). Why does it control the translation of JAK2 and ATF4 but not GAPDH? This point should be at least discussed in the text.
4. In Fig 2E-F, please specify the dataset reference(s) used in the GSEA plot and indicate DENR WT vs. KO or DENR KO vs. WT in the x-axis.
5. The authors did not mention the criteria of the high and low expression of DENR in Fig 7E-G. Is high or low expression compared with the median or the average value? Please clarify it.
6. The protein markers should be added in each of the blots.

Reviewer #4:

Remarks to the Author:

The manuscript describes a focused CRISPR screen targeting all RBPs to find regulators of IFN γ mediated PD-L1 expression. DENR was identified to regulate PD-L1 expression through Jak2. Overall the manuscript has merit, but more experimental data is required to fully underpin the hypothesis of their work.

While I have some minor comments and questions on some experimental setups, I would like to suggest some changes in the functional experiments of tumour cell growth and killing.

- 1, In Figure 6 the authors look at the growth of DENR KO cells in mice. Overall their conclusion is that the tumours are eradicated much better by the action of CD8 T cells. However, it is already apparent that even the dtTomatoe high expressing cells are depleted as shown in Figure 6b. Could the authors please comment on this?
- 2, In the following Figures they show that the overall tumour growth of DENR KO cells is slower? However, is this due to PD-L1 expression or could it also be that the cells are not as fit as the DENR wt cells (e.g. MC38 in Figure 6J)? To convincingly show at least an involvement of CD8 T cells in the slowing down of the tumour cells, the authors could delete CD8 T cells by administering anti-CD8 antibodies to the mice and then transplant the DENR KO and wt tumours. Should this not be within "reach" for the authors, transplantation into Rag1 deficient mice would do as well. The expectation would be that the wt and DENR KO tumours grow at a similar pace.
- 3, In Figure 7a-d the authors show the direct killing of OT-1 CD8 T cells of DENR wt and ko MC38-OVA tumours. I assume the PD-L1 upregulation in the tumours caused by CD8 is through IFN γ ? Could the authors demonstrate by IC flow or western blot of the tumour cells that indeed the IFN pathway is switched on in response to the CD8T cell incubation?
- 4, The increase in apoptosis should be accompanied by a increase in cleaved caspase3. Could the authors please analyse by intracellular flow or western that caspase-3 is indeed increased upon CD8 T cell incubation in Figure 7D? The killing seems to be in general very weak. Would higher effector to target ratio or longer incubation increase the effects to make the point much stronger? Do the authors have evidence that the CD8 mediated killing is indeed dependent on OVA expression in the tumour cells? This could be tested by incubating the CD8 T cells with MC38 cells without expressing OVA.

Minor points:

1, All the western blots should be labelled with size markers and densitometric quantified as some of the differences are relatively small.

2, Could the authors elaborate a little more on the sgRNA library. How did they design the sgRNAs?

3, While hits are clearly identified in the screen, it would be nice to see the IFN γ induced PD-L1 expression in Figure 1 without sgRNA library. Will there be still some low and high expressors?

4, I don't quite understand the experimental setup for DENR OE using sgRNA against DENR in Ext Fig2b? Could the authors explain this.

5, In Figure 4g it would be important to show that the differences in GFP are not due to differences in GFP expression, but the effects of the uORFs and DENR. Could the authors please analyse the RNA levels of GFP in all experimental groups? There shouldn't be any differences on RNA levels of GFP.

6, The polysome analysis from the HELA cells is nice. However, I am not quite sure why the polysomes seem to stall in different positions depending on the DENR KO, i.e. KO1 different to KO2. Could the authors please explain this observation?

Response to the Reviewers:

Reviewer #1 (Remarks to the Author): with expertise in translational control

The paper is addressing an interesting question in determining the patients that could benefit from anti-PD-1/PD-L1 blocked and unravelling the molecular mechanism would be important. However, there are many inconsistencies in the work and the authors do not really address the question of why patients with high DENR are less sensitive to PD-L1 blockade. In addition, the paper is rather rambling and would benefit from some of the figures, which essentially show negative data, being moved to the supplementary section to provide a more focused story.

We thank the reviewer for his/her enthusiastic support and critical evaluation of our findings and manuscript.

In our study, we have identified that DENR positively regulates PD-L1 expression through IFN γ -JAK2-STAT1 signaling, DENR high patients is LESS sensitive to PD-L1/PDL1 immune check blockade (ICB) therapy. The results seem inconsistent to some previous study showing some PD-L1 high patients were more sensitive to PD-L1/PDL1 blockade. However, PD-L1/PDL1 blockade therapy only benefit ~20% cancer patients, and among the remaining 80% patients, many of them have mid-high PD-L1 expression¹. In Tumor Microenvironment, a major source of IFN γ is T cells. Infiltrating T cells secrete more IFN γ , which stimulate tumor cells via JAK-STAT signaling to overexpress PD-L1, as well as many other known and unknown molecules, to inhibit T cell function². This could explain that only ~20% patients could benefit from ICB. It has been well studied that interfering with tumor IFN-JAK-STAT signaling combined with PD-1/PD-L1 immune check blockade (ICB) monotherapy was more effective than only ICB monotherapy². Our current study proves that DENR directly regulates JAK2, and downregulation of DENR will inhibit IFN-JAK-STAT signaling and its downstream PD-L1 and many other IFN response genes. Thus, DENR low patients with weakened JAK-STAT signaling could be more sensitive to ICB therapy. Multiple previous studies also have reported that candidate genes positively regulate tumor expression of PD-L1 is associated with PD-1 blockade resistance^{3,4}. However, the clinical data is limited in our study, and the DENR function in tumor microenvironment of the tumor patients is very complex and is in need of further investigation. This point now has been included in our discussion.

As the Reviewer suggested, some of the figures with non-essential negative data are now moved to the supplementary section, and some are re-organized to provide a more focused story in the revised manuscript.

Major comments

1. Regarding the statistical analysis in Figures 1G, 1I, 2B, 3H, 3G and 7B - multiple comparisons (multiple t-tests) have been made between the data in each figure panel. Running multiple t-tests in this manner increases the chance of error, therefore, statistical analysis should take these multiple comparisons into account by using ANOVA with Dunnett's post hoc test.

We thank the reviewer for this valuable comment. As suggested, ANOVA with Dunnett's post hoc test has been used in multiple comparisons, as shown in Fig. 1e, 2b, 2c, 3c, 3e and Supplementary Fig. 2b, 3d.

2. In the introduction the authors explain that IFN gamma is a known activator of the JAK-STAT pathway, yet Fig 3a shows that IFN gamma does not phosphorylate JAK2 in control sgRNA cells (and Fig 3b shows no phosphorylation of JAK1 with IFN gamma). Can the authors explain why DENR KO cells behave this way? Additionally, densitometry (for phospho proteins normalised to the total level of the protein) need to be carried out and statistical analysis of the western blot bands is required as some changes are very subtle.

We thank the reviewer for bringing our attention to this very important point which indeed greatly improve the quality of our study. IFN γ is a strong well-known activator of the JAK-STAT pathway, we used it to induce PD-L1 expression, under which condition we could observe dramatic reduction of PD-L1 in DENR KO cells. In our first version of manuscript, we only paid attention to the total protein levels and found JAK2 was reduced in DENR KO cells, however, as the reviewer correctly pointed out, the p-JAK2 as well as p-JAK1 was not obviously increased even in WT control cells after treated with IFN γ for 1 h, the time point when we could readily detect the dramatically decreased p-STAT1/3 protein levels.

To address the problem, we then research the literatures which suggest that the phosphorylation of JAK1 and JAK2 is a quick response upon stimulation with high Western Blotting background. To achieve low background blot, the cells need to be cultured in serum-free medium for overnight before cytokine stimulation⁵. According to this, we cultured the cells without FBS overnight before treated with IFN γ for only 15 to 30 min for WB, we then observed increased phosphorylation levels of JAK1 and JAK2 at 15min and 30 min, as shown in Fig. 3a and Supplementary Fig. 2c. As the total protein levels of JAK2 are also changed in DENR KO cells, the density of p-JAK2 and JAK2 is normalized to GAPDH, and the quantification is shown in Fig. 3b. The results show that DENR depletion significantly reduce the protein levels of JAK2/p-JAK2 but not JAK1/p-JAK1.

3. Figure 4. The appears to be almost no variation on the bar charts shown for figure 4G which surprising. Are these technical repeats or biological repeats?

We thank the reviewer for this valuable comment. The data shown in Fig. 4f are technical repeats. We also performed three biological repeats, which showed similar conclusion. Due to variation of the transient transfection efficiency, when combining the biological repeats, the data variation is big.

Comparing to Control sgRNA transfected cells, DENR sgRNA depletion just reduce 15% EGFP+ cells with the 1 uORF reporter, suggesting that the 1 uORF is not a strong element for DENR's function

and this is consistent with other study⁶. In the revised manuscript, we repeated again this GFP reporter experiment and the corresponding mRNA levels of different GFP reports were measured as shown in Fig. 4g, indicating that the DENR regulates GFP expression at the translation level via uORF elements.

4 Related to Figure 5c: the author's state "Together, these data exclude the possibility that DENR has a broad effect on the translation of all mRNAs...". However, polysome profiling does not clearly show changes in translation elongation. For example, stalled (non-elongating) ribosomes would remain bound to RNA and appear as heavy polysomes, which would be interpreted as active translation. Moreover, there is a decrease in 80S in the DENR KO cells, but no corresponding change in 40S or 60S or heavy polysomes. This raises a number of questions, such as where are the free 80S ribosomes going? Are they being degraded or are they being recruited to insoluble granules? if either of these were a possibility then translation rates would most likely be decreased.

Therefore, to validate the statement and exclude that DENR has a broad effect on translation of all mRNA the authors should also:

- Investigate the rate of translation initiation (i.e. with radiolabelled methionine incorporation)
- Determine the phosphorylation status of translation factors that are known to inhibit protein synthesis (eIF2a, 4EBP1, eEF2) in the control and KO cell lines.
- Although ribosome profiling of other studies has been analysed it is important to carry out RT-qPCR on polysome fractions for individual mRNAs of interest (such as *Jak1/2*) to determine if they are associated with heavy polysomes (i.e. being actively translated) in this cell line KO model.

We thank the reviewer for the very constructive suggestions. As suggested, we performed the following three experiments to strength our statements.

(1) To exclude that DENR has a broad effect on translation, we used the Click-iT[®] Plus OPP Protein Synthesis Assays (ThermoFisher) to detect newly synthesized proteins. This reagent is a puromycin analog containing an alkyne moiety, which is a non-radioactive alternative to radiolabelled methionine incorporation. When added to culture media, OPP is readily taken up by actively growing cells. OPP inhibits protein synthesis by disrupting peptide transfer on ribosomes causing premature chain termination during translation. Addition of the Alexa Fluor[®] picolyl azide and the Click reaction reagents leads to a chemoselective ligation or "click" reaction between the picolyl azide dye and the alkyne OPP, allowing the modified proteins to be detected by flow cytometry. Oligomycin A (a mitochondrial FOF1-ATPase inhibitor) was added in WT cells as a control to delay the protein synthesis with reduced fluorescence as detected. Following the protocol, our results showed that knocking out DENR did not reduce the fluorescence intensities comparing to Control cells, suggesting that DENR KO does not have a broad effect on translation (Supplementary Fig. 3c and d).

We analyzed the phosphorylation and total protein levels of eIF2a, 4EBP1, and eEF2 by WB, and no changes were observed in Control and DENR KO cells (Supplementary Fig. 3a).

In addition, we carried out RT-qPCR on each polysome fraction in Control and DENR KO cells, and the results showed that *Jak1* mRNAs without uORFs did not have any significant ribosome association

changes with each fractions (**Fig. 5d**), however, the distribution of Jak2 mRNAs displayed a leftward shift on the fractionation gradient, indicating that *Jak2* mRNAs associated with smaller polysomes upon DENR depletion (**Fig. 5e**), suggesting that the translation efficiency of JAK2 but not JAK1 were suppressed in DENR KO cells.

5. There are inconsistencies in the way PD-L1 levels are shown. Some figures use only total PD-L1 in western blots (i.e. Fig2A), whereas others quantify surface PD-L1 (i.e. Fig 1). Why is cell surface quantification used in some experiments and not others? Is there a population of PD-L1 in the cell that is not presented on the cell surface that would be detected in the western blot? It is important to include both types of analysis throughout the manuscript.

We thank the reviewer for bringing our attention to this point. PD-L1 is mainly a surface protein and we often use flow cytometry to quantify its protein levels for convenience as we are a professional immunology lab.

As the reviewer suggested, we now re-run all the related experiment by both WB and FACS, and the data is shown in **Fig. 1d and e**, **Fig. 2a and b**, **Fig. 3d and e**, and **Supplementary Fig. 2a and b**.

6. Jak1 mRNA is also bound by DENR, yet DENR KO does not have any impact on JAK1 protein or mRNA levels. Could the authors expand on why this may be the case?

We thank the reviewer for this valuable comment. DENR does not show specific binding motif and the DENR crystal structure shows that it is a part of the 40S ribosome⁷. Thus, binding of DENR on mRNAs may depend on ribosomes, which could be the explanation that we could also observe DENR bound to *Jak1* mRNA without uORFs. As a translation re-initiation factor, DENR helps ribosomes to overcome uORFs only when they are present in 5'UTRs of certain mRNAs. Thus, DENR binds mRNA without specificity, but has functional specificity.

To prove its functional specificity, we carried out RT-qPCR on each polysome fraction in Control and DENR KO cells, and the results showed that Jak1 mRNAs without uORFs did not have any significant ribosome association changes with each fractions (**Fig. 5d**), however, the distribution of *Jak2* mRNAs displayed a leftward shift on the fractionation gradient, indicating that *Jak2* mRNAs associated with smaller polysomes upon DENR depletion (**Fig. 5e**), suggesting that the translation efficiency of JAK2 but not JAK1 were suppressed in DENR KO cells. In addition, we also reanalyzed a ribosome footprint sequencing data set (GSE140084) in DENR KO versus WT HeLa cells and found that loss of DENR led to the accumulation of 40S and 80S on uORFs in JAK2 (**Fig. 5f**) but not JAK1 (**Supplementary Fig. 3b**).

Minor comments

- Extended figure 2 – no key for colours of bars.
- Figure 4F and 4G – the labelling of uORFs for 4F is actually part of panel 4g and so is unclear.
- Figure 6J and 6K are referenced in the text before 6I.
- Please add a label to distinguish the studies illustrated in figure 7E and 7F.
- Figure 7H is not referenced in the main text.

We apologize for those errors and really appreciate the reviewer's efforts to improve the manuscript.

In the revised manuscript, the key for colors of bars were added in Supplementary Fig. 2f. The labelling of uORFs is now clearly shown in Fig. 4e and f.

Fig. 6f and g are referenced in the text after Fig. 6d and e. Labels are added to distinguish the studies illustrated in Fig. 7d-f, and Fig. 7g is referenced in the main text now.

Reviewer #2 (Remarks to the Author): with expertise in JAK/STAT pathway, cancer

This manuscript provides evidence supporting the notion that an RNA-binding protein (RBP), density regulated re-initiation and release factor (DENR), regulates PD-L1 expression by controlling the translation of JAK2, a component of the upstream signaling pathway known to regulate PD-L1. The authors identified DENR in a targeted CRISPR/Cas9 screen for RBPs that may influence PD-L1 expression, using a pool of sgRNAs targeting 1467 RBPs. They found that DENR depletion reduced PD-L1 expression in stable Cas9-expressing RAW264.7 cells after IFN γ treatment. They further determined that JAK2 protein translation is specifically reduced in DENR depleted cells, and that 3 uORFs present in the 5' region of JAK2 mRNA mediate the effect.

We thank the reviewer for his/her enthusiastic support of our findings and manuscript. The reviewer accurately summarizes our key findings in detail.

A major concern is that DENR as a general RBP should have broad targets and is unlikely a specific regulator of JAK2, much less PD-L1. Depleting DENR may have many unintended off-targets, thus reducing its potential as a therapeutic target for immunotherapy.

We thank the reviewer for her/his critical evaluation of our work. As suggested, we analyzed the global genes and identified 2635 genes with at least 1 uORF. We then overlap those genes with translation deficiency genes defined from published ribosome profiling sequencing data set of DENR KO NIH3T3 cells (GSE116221), the results showed there are 106 genes with at least 1 uORF and decreased translation efficiency, and Jak2 were among them (Fig. 4h). We agree with the reviewers that DENR as a translational regulator has many other targets in addition to JAK2. However, based on our data, JAK2 is

one of DENR major targets that mediate the tumor immune evasion through PD-L1. Depleting DENR may have many unintended off-targets, but we showed with solid evidences that tumor growth was significant inhibited after DENR depletion, due to enhanced CD8+ T cell infiltration and killing. DENR as a therapeutic target for immunotherapy does need further investigation.

Other concerns are as the following.

Methods: Need to provide the design and sequence information about the sgRNAs used in the screen and specific knock down experiments.

We thank the reviewer for bringing our attention to this point. The design of sgRNA library for screen was added in the method section of the revised manuscript, and the sequence information was provided in Supplementary Table 4. In detail, the sgRNA library was designed by the website <http://cistrome.org/crispr-focus/>. On the website, we upload the gene list (RBPs and positive regulators) and selected the organism, number of sgRNAs for each gene and number of Control sgRNAs that we could get the sgRNA library. The sgRNAs used in the screen and the sgRNA sequence data was uploaded in supplementary material or GEO (GSE184048). The sgRNAs used for specific knock down were list in the Supplementary Table 2.

IFN γ treatment: “20 ng/ml treatment for 4 h” in Methods for screen, but “for 24 h” in Extended Fig. 2A. In other places, no specifics were mentioned. Need to provide details and rationale. Results for 4 h and 24 h treatment should be very different on JAK/STAT signaling.

We thank the reviewer for this great suggestion and apologize for the confusion caused by our negligence. For the sensitivity of CRISPR screening, the PD-L1 should be expressed at a moderate level as some studies suggested⁸. We had then tested different IFN γ treatment conditions and time point, and found out that PD-L1 expression on RAW cells peaked around 8 h, and reached a moderate level at 4 h that is suitable for screening (Supplementary Fig. 1a). Thus, cells were treated for 4 h under 20 ng/ml IFN γ treatment for CRISPR screen.

As the phosphorylation of JAK/STAT is a quick response. In Fig 2g, to measure the p-STAT1/3, cells were treated for 1 h. In Fig. 3a, to detect p-JAK1/p-JAK2, cells were treated for 15 or 30 min as studies suggested⁵. In other figures, the total or surface PD-L1 were measured by WB and FACS 24 h after stimulation to allow PD-L1 full expression. The details is now described in the revised manuscript.

Fig. 1. DENR knockout (KO): How were the KO cells created? Were they stable lines or pooled polyclonal cells? If stable lines, need to provide sequence information regarding genomic deletions. Should provide sequence information for denr sgRNA1-3.

We thank the reviewer for bringing our attention to this point, The KO cell lines were created by stably expressing Cas9 and corresponding specific sgRNAs in pooled polyclonal cells, the sgRNA sequence information of DENR and other targets (JAK1/2, STAT1/3, PD-L1) were list in the Supplementary Table 2.

Do JAK and STAT KO affect PD-L1 expression? How do they compare with denr KO?

We appreciate the reviewer for raising this very constructive point. As expected, JAK1, JAK2 and STAT1, but not STAT3, were also identified as positive regulators of PD-L1 in our screen (Fig. 1c). As suggested, we newly created JAK1, JAK2, STAT1, STAT3 and PD-L1 knock out MC38 cell lines, the total and surface PD-L1 were measured by WB and quantitated by FACS. We found that knock out JAK1, JAK2, STAT1 and DENR, but not STAT3, could dramatically suppressed PD-L1 expression at similar levels (Fig. 3d and e).

Fig. 2. Not clear what was done in “partly rescued by the transient overexpression of an sgRNA against DENR (Extended Data Fig. 2b), which excluded the off-targeting of JAK2 by DENR sgRNAs.” Different sgRNAs or JAK2 overexpression? What exactly were overexpressed?

We really apologize for the confusion caused by our negligence. This is a wrong wording of “against”. We actually mean to overexpress a “sgRNA-resistant DENR cDNA expression vector”. To exclude the off-target of DENR sgRNAs, DENR cDNA need to be overexpressed in DENR KO cells to verify whether it could rescue JAK2 expression. As DENR KO cells were created by stably expressing of Cas9 and DENR sgRNA, thus the continuously expressed DENR sgRNAs will also target and cut the transfected WT DENR cDNAs. Thus, as some studies suggested, we switched the sequence at the DENR sgRNA target area with synonymous codons. The sgRNA-resistant DENR variant could be translated into the same WT DENR protein (Supplementary Fig. 2e), but is resistant to the cut of DENR sgRNAs. We named it “sgRNA-resistant DENR cDNA” in the revised manuscript.

JAK2 IP: Did the IP pull down any JAK2 interactors? How do we know the IP worked?

We thank the reviewer for bringing our attention to this point. STAT3 is a strong interactor with JAKs⁹. In our study, JAK2 IP could also pull down STAT3, but not DENR, as shown in (Supplementary Fig. 3a).

Fig.4. Should have systematic scan for the presence of uORFs in genes that are affected by DENR knock down vs. those not affected. It is unlikely that JAK2 is the only target regulated by DENR.

We thank the reviewer for her/his critical evaluation of our work. As suggested, we analyzed the global genes and identified 2635 genes with at least 1 uORF. We then overlap those genes with translation deficiency genes defined from published ribosome profiling sequencing data set of DENR KO NIH3T3 cells (GSE116221), the results showed there are 106 genes with at least 1 uORF and decreased translation efficiency, and *Jak2* were among them (Fig. 4h). We agree with the reviewers that DENR as a translational regulator has many other targets in addition to JAK2. However, based on our data, JAK2 is one of DENR major targets that mediate the tumor immune evasion through PD-L1.

Fig. 5. Need to explain/analyze why DENR associated with GAPDH without affecting its expression. Any uORFs present in *Gapdh* transcript?

We thank the reviewer for this valuable comment. DENR does not show specific binding motif and the DENR crystal structure shows that it is a part of the 40S ribosome⁷. Thus, binding of DENR on mRNAs may depend on ribosomes, which could be the explanation that we could also observe DENR bound to *Jak1* and *Gapdh* mRNAs without uORFs. As a translation re-initiation factor, DENR helps ribosomes to overcome uORFs only when they are present in 5'UTRs of certain mRNAs. Thus, DENR binds mRNA without specificity, but has functional specificity.

To prove its functional specificity, we carried out RT-qPCR on each polysome fraction in Control and DENR KO cells, and the results showed that *Jak1* mRNAs without uORFs did not have any significant ribosome association changes with each fractions (Fig. 5d), however, the distribution of *Jak2* mRNAs displayed a leftward shift on the fractionation gradient, indicating that *Jak2* mRNAs associated with smaller polysomes upon DENR depletion (Fig. 5e), suggesting that the translation efficiency of JAK2 but not JAK1 were suppressed in DENR KO cells. In addition, we also reanalyzed a ribosome footprint sequencing data set (GSE140084) in DENR KO versus WT HeLa cells and found that loss of DENR led to the accumulation of 40S and 80S on uORFs in JAK2 (Fig. 5f) but not JAK1 (Supplementary Fig. 3b).

Reviewer #3 (Remarks to the Author): with expertise in cancer immunology

In this study, the authors have reported that an RNA binding protein, DENR, attenuates IFN γ -induced expression of PD-L1 by inhibiting translation of JAK2 mRNA. They have shown that DENR directly binds to the uORFs in the 5'UTR of JAK2 mRNA, and thereby helps the 40S ribosomes to bypass the uORFs and initiate translation of JAK2. Consequently, the upregulation of JAK2 mediates IFN γ -induced expression of PD-L1 to evade the killing of host anti-tumor CD8⁺ T cells. These findings are well-designed and precise. However, there are some minor issues that should be addressed before its

acceptance by Nat Commun.

We thank the reviewer for his/her enthusiastic support of our findings and manuscript. The reviewer accurately summarizes our key findings in detail and highlights the strength of our work.

1. As DENR binds to uORFs to help mRNA translation, it is critical to examine whether knockout of DENR has any effects on tumor cell growth or death in cultures and in immune-compromised mice.

We sincerely thank the Reviewer for the valuable and insightful comments. In the revised manuscript, we now provide new data showing that knocking out DENR did not impair the global protein synthesis as detected by a “Click-OPP” nascent protein translation assay (Supplementary Fig. 3c and d), did not inhibit in vitro cell growth as detected by CCK8 cell proliferation assay (Supplementary Fig. 5a and b), did not induce apoptosis and cell death (E : T=0:1 in Fig. 7a and b , Supplementary Fig. 5e and f). In addition, as the reviewer kindly suggested, we transplanted DENR KO and Control cells to RAG1^{-/-} mice and observed that DENR KO B16/F10 tumors, as well as DENR KO MC38 tumors to a lesser extent, could grow at a similar pace comparing to Control tumors (Supplementary Fig. 5c and d). Thus, the observed tumor growth inhibition after DENR depletion in the syngeneic tumor models is mainly due to enhanced CD8⁺ T cell killing in the TME, other than intrinsic tumor cell growth defects.

2. In Fig 7C-E, the authors detected the apoptosis rate to assess the killing of tumor cells by CD8 + T cells. The difference is rather minor. CD8⁺ T cells-mediated tumor killing also induces necroptosis, ferroptosis and pyroptosis. It would be more helpful if the authors use a board cell-death measurement to assess the ratio of cell death.

We truly appreciate this insightful advice. In the revised manuscript, we performed both the global cell death assay (by LIVE/DEAD™ Fixable Near-IR Dead Cell Stain Kit from Thermo Fisher) and a different apoptosis assay (by CellEvent™ Caspase-3/7 Green kit from Thermo Fisher). The apoptosis ratio at E : T (effector : tumor) =3:1 was still minor, similar to the results measured previously by Annexin V and 7-AAD in Control + OVA and DENR KO + OVA cells. However, we observed dramatically increased apoptosis rate and global cell death in DENR KO + OVA cells comparing to Control + OVA cells at the E : T= 5:1 (Fig. 7a and b , Supplementary Fig. 5e and f).

3. DENR binds to mRNAs of different genes (for example, JAK2, ATF4 and GAPDH in this study). Why does it control the translation of JAK2 and ATF4 but not GAPDH? This point should be at least discussed in the text.

We thank the reviewer for this valuable comment. DENR does not show specific binding motif and the DENR crystal structure shows that it is a part of the 40S ribosome⁷. Thus, binding of DENR on mRNAs may depend on ribosomes, which could be the explanation that we could also observe DENR bound to *Jak1* and *Gapdh* mRNAs without uORFs. As a translation re-initiation factor, DENR helps ribosomes to overcome uORFs only when they are present in 5'UTRs of certain mRNAs. Thus, DENR binds mRNA without specificity, but has functional specificity.

To prove its functional specificity, we carried out RT-qPCR on each polysome fraction in Control and DENR KO cells, and the results showed that *Jak1* mRNAs without uORFs did not have any significant ribosome association changes with each fractions (**Fig. 5d**), however, the distribution of *Jak2* mRNAs displayed a leftward shift on the fractionation gradient, indicating that *Jak2* mRNAs associated with smaller polysomes upon DENR depletion (**Fig. 5e**), suggesting that the translation efficiency of JAK2 but not JAK1 were suppressed in DENR KO cells. In addition, we also reanalyzed a ribosome footprint sequencing data set (GSE140084) in DENR KO versus WT HeLa cells and found that loss of DENR led to the accumulation of 40S and 80S on uORFs in JAK2 (**Fig. 5f**) but not JAK1 (**Supplementary Fig. 3b**).

4. In Fig 2E-F, please specify the dataset reference(s) used in the GSEA plot and indicate DENR WT vs. KO or DENR KO vs. WT in the x-axis.

We thank the reviewer for this comment. In **Fig. 2f** the gene set “c2.all.v7.5” for GSEA analysis was downloaded from the MSigDB database and was noted in the method section in the revised manuscript. “DENR KO” and “Control” were indicated in the x-axis.

5. The authors did not mention the criteria of the high and low expression of DENR in Fig 7E-G. Is high or low expression compared with the median or the average value? Please clarify it.

We thank the reviewer for bringing our attention to this point. The method to analyze the survival probability was now included in the method section of the revised manuscript. In brief, survminer package was used to determine the cutoff point of survival information for each dataset based on the association between DENR expression and patient overall survival. To find the maximum rank statistics and reduce the calculated batch effect, the "surv-cutpoint" function was used to dichotomy DENR expression and all potential cutting points were repeatedly tested, then the patient samples were divided into the high-DENR group and the low-DENR group **according to the maximum selected log-rank statistics**. Kaplan-Meier comparative survival analyses for prognostic analysis were generated, and the log-rank test was used to determine the significance of the differences. All statistical analysis was two-side and considered $p < 0.05$ as statistical significance.

6. The protein markers should be added in each of the blots.

We thank the reviewer for bringing our attention to this point, the protein markers are now added throughout the manuscript.

Reviewer #4 (Remarks to the Author): with expertise in CRISPR screens, immunology, cancer

The manuscript describes a focused CRISPR screen targeting all RBPs to find regulators of IFN γ mediated PD-L1 expression. DENR was identified to regulate PD-L1 expression through Jak2. Overall the manuscript has merit, but more experimental data is required to fully underpin the hypothesis of their work.

While I have some minor comments and questions on some experimental setups, I would like to suggest some changes in the functional experiments of tumour cell growth and killing.

We thank the reviewer for his/her enthusiastic support of our findings, and appreciate her/his efforts made to help us to improve the quality of our manuscript.

1, In Figure 6 the authors look at the growth of DENR KO cells in mice. Overall their conclusion is that the tumours are eradicated much better by the action of CD8 T cells. However, it is already apparent that even the dtTomato high expressing cells are depleted as shown in Figure 6b. Could the authors please comment on this?

We sincerely thank the Reviewer for the valuable comments. In **Fig. 6a and b** we mixed equivalent amount of DENR KO (EGFP) and Control (tdTomato) cells and subcutaneously implanted into the mice. As the reviewer pointed, the FACS shown that DENR KO (EGFP) cells were eradicated, however, Control (tdTomato) cells were also partly depleted. We think that DENR KO cells with reduced PD-L1 expression in the tumor result in more CD8 $^+$ T infiltrating and killing, the increased CD8 $^+$ T cells in the TME will not only kill DENR KO cells, but also kill Control (tdTomato) cells with less efficiency. Also as shown in the in vitro killing assay (**Fig. 7a and b**), CD8 $^+$ T cells have better killing ability against DENR KO cells, but increased CD8 $^+$ T cells / tumor cells (E : T) ratio will also lead to enhanced apoptosis rates of WT control cells.

2, In the following Figures they show that the overall tumour growth of DENR KO cells is slower? However, is this due to PD-L1 expression or could it also be that the cells are not as fit as the DENR wt cells (e.g. MC38 in Figure 6J)? To convincingly show at least an involvement of CD8 T cells in the

slowing down of the tumour cells, the authors could delete CD8 T cells by administering anti-CD8 antibodies to the mice and then transplant the DENR KO and wt tumours. Should this not be within "reach" for the authors, transplantation into Rag1 deficient mice would do as well. The expectation would be that the wt and DENR KO tumours grow at a similar pace.

We sincerely thank the Reviewer for the valuable and insightful comments. In the revised manuscript, we now provide new data showing that knocking out DENR did not impair the global protein synthesis as detected by a "Click-OPP" nascent protein translation assay (Supplementary Fig. 3c and d), did not inhibit in vitro cell growth as detected by CCK8 cell proliferation assay (Supplementary Fig. 5a and b), did not induce apoptosis and cell death (E : T=0:1 in Fig. 7a and b ,Supplementary Fig. 5e and f). In addition, as the reviewer kindly suggested, we transplanted DENR KO and Control cells to RAG1^{-/-} mice and observed that DENR KO B16/F10 tumors, as well as DENR KO MC38 tumors to a lesser extent, could grow at a similar pace comparing to Control tumors (Supplementary Fig. 5c and d). Thus, the observed tumor growth inhibition after DENR depletion in the syngeneic tumor models is mainly due to enhanced CD8⁺ T cell killing in the TME, other than intrinsic tumor cell growth defects.

3, In Figure 7a-d the authors show the direct killing of OT-1 CD8 T cells of DENR wt and ko MC38-OVA tumours. I assume the PD-L1 upregulation in the tumours caused by CD8 is through IFN γ ? Could the authors demonstrate by IC flow or western blot of the tumour cells that indeed the IFN pathway is switched on in response to the CD8T cell incubation?

We are very grateful for the great suggestion. The p-STAT1 was measured by IC flow as shown in Supplementary Fig. 5g, we observed that knocking out DENR reduced p-STAT1 comparing to Control cells, indicating the IFN pathway is switched on in response to the CD8⁺ T cell in co-culture.

4. The increase in apoptosis should be accompanied by a increase in cleaved caspase3. Could the authors please analyse by intracellular flow or western that caspase-3 is indeed increased upon CD8 T cell incubation in Figure 7D? The killing seems to be in general very weak. Would higher effector to target ratio or longer incubation increase the effects to make the point much stronger? Do the authors have evidence that the CD8 mediated killing is indeed dependent on OVA expression in the tumour cells? This could be tested by incubating the CD8 T cells with MC38 cells without expressing OVA.

We truly appreciate this insightful advice. In the revised manuscript, we performed both the global cell death assay (by LIVE/DEAD™ Fixable Near-IR Dead Cell Stain Kit from Thermo Fisher) and a different apoptosis assay (by CellEvent™ Caspase-3/7 Green kit from Thermo Fisher). As suggested, we now used OVA expressing MC38 tumor cells and OT-1 T cells for co-culture. The apoptosis ratio at E:T (effector : tumor) =3:1 was still minor, similar to the results measured previously by Annexin V and

7-AAD in Control + OVA and DENR KO + OVA cells. However, we observed dramatically increased apoptosis rate and global cell death in DENR KO + OVA cells comparing to Control + OVA cells at the E:T= 5:1, and the CD8+ T killing is dependent on OVA expression in the tumor cells. (**Fig. 7a and b**, **Supplementary Fig. 5e and f**).

Minor points:

1. All the western blots should be labelled with size markers and densitometric quantified as some of the differences are relatively small.

We thank the reviewer for this advice. We labelled size markers with all the western blots, and quantified in some figures when the differences are small as in **Fig. 3a, Supplementary Fig. 2d and e**.

2. Could the authors elaborate a little more on the sgRNA library. How did they design the sgRNAs?

We thank the reviewer for this advice. A detail description of the sgRNA library was included in the method section in the revised manuscript. The sgRNA library was designed by the website <http://cistrome.org/crispr-focus/>. On the website, we uploaded the gene list (RBPs and positive regulators) and selected the organism, number of sgRNAs for each gene and number of Control sgRNAs that we could get the sgRNA library. The full list of sgRNA library and sequence data was also included in supplementary material, **Supplementary table 4** and GEO (GSE184048).

3. While hits are clearly identified in the screen, it would be nice to see the IFN γ induced PD-L1 expression in Figure 1 without sgRNA library. Will there be still some low and high expressors?

We are very grateful for the kind suggestion. While we did the control experiment, we did not show IFN γ induced PD-L1 expression in RAW cells without sgRNA library in our first submitted manuscript. As shown in **Supplementary Fig. 1c**, we did not observe marked different change of PD-L1 low or high expressors.

4. I don't quite understand the experimental setup for DENR OE using sgRNA against DENR in Ext Fig2b? Could the authors explain this.

We really apologize for the confusion caused by our negligence. This is a wrong wording of “against”. We actually mean to overexpress a “sgRNA-resistant DENR cDNA expression vector”. To exclude the off-target of DENR sgRNAs, DENR cDNA need to be overexpressed in DENR KO cells to

verify whether it could rescue JAK2 expression. As DENR KO cells were created by stably expressing of Cas9 and DENR sgRNA, thus the continuously expressed DENR sgRNAs will also target and cut the transfected WT DENR cDNAs. Thus, as some studies suggested, we switched the sequence at the DENR sgRNA target area with synonymous codons. The sgRNA-resistant DENR variant could be translated into the same WT DENR protein (Supplementary Fig. 2e), but is resistant to the cut of DENR sgRNAs. We named it “sgRNA-resistant DENR cDNA” in the revised manuscript.

5. In Figure 4g it would be important to show that the differences in GFP are not due to differences in GFP expression, but the effects of the uORFs and DENR. Could the authors please analyse the RNA levels of GFP in all experimental groups? There shouldn't be any differences on RNA levels of GFP.

We are very grateful for the constructive suggestion. We repeated this experiment and the corresponding mRNA levels of GFP were measured, no differences were observed between DENR KO and Control cells or all experimental groups, suggesting that the difference in %EGFP+ cells are not due to differences in GFP transcription, but due to the translation regulation by DENR (Fig. 4h).

6. The polysome analysis from the HELA cells is nice. However, I am not quite sure why the polysomes seem to stall in different positions depending on the DENR KO, i.e. KO1 different to KO2. Could the authors please explain this observation?

We really appreciate the reviewer for raising this great comment. In our first submitted manuscript, the position difference of polysome peaks between DENR KO1 and KO2 is due to our inexperienced bioinformatic analysis. With help from professional bioinformatician from our collaborator's lab, we now re-do the polysome analysis by firstly referencing the Ribo-data to corresponding RNA-seq data, to reduce the effects of sample preparation on data analysis. As shown in Fig. 5f and Supplementary Fig. 3b, loss of DENR led to the accumulation of 40S and 80S on uORFs in JAK2 but not JAK1.

References

1. Sun, C., Mezzadra, R. & Schumacher, T. N. Regulation and Function of the PD-L1 Checkpoint. *Immunity (Cambridge, Mass.)*. **48**, 434-452 (2018).
2. Benci, J. L. et al. Tumor Interferon Signaling Regulates a Multigenic Resistance Program to Immune Checkpoint Blockade. *Cell*. **167**, 1540-1554 (2016).
3. Terry, S. et al. Association of AXL and PD-L1 Expression with Clinical Outcomes in Patients with Advanced Renal Cell Carcinoma Treated with PD-1 Blockade. *Clin. Cancer Res.* **27**, 6749-6760 (2021).
4. Deng, Y. et al. Glucocorticoid receptor regulates PD-L1 and MHC-I in pancreatic cancer cells to promote immune

evasion and immunotherapy resistance. *Nat. Commun.* **12**, (2021).

5. Liu, A., Liu, Y., Li, P. K., Li, C. & Lin, J. LLL12 inhibits endogenous and exogenous interleukin-6-induced STAT3 phosphorylation in human pancreatic cancer cells. *Anticancer Res.* **31**, 2029-2035 (2011).
6. Schleich, S., Acevedo, J. M., Clemm Von Hohenberg, K. & Teleman, A. A. Identification of transcripts with short stuORFs as targets for DENR•MCTS1-dependent translation in human cells. *Sci. Rep.-UK.* **7**, (2017).
7. Lomakin, I. B., De, S., Wang, J., Borkar, A. N. & Steitz, T. A. Crystal structure of the C-terminal domain of DENR. *Computational and Structural Biotechnology Journal.* **18**, 696-704 (2020).
8. Mezzadra, R. et al. Identification of CMTM6 and CMTM4 as PD-L1 protein regulators. *Nature.* **549**, 106-110 (2017).
9. Kim, B. et al. NSC114792, a novel small molecule identified through structure-based computational database screening, selectively inhibits JAK3. *Mol. Cancer.* **9**, (2010).

Reviewers' Comments:

Reviewer #1:

Remarks to the Author:

The authors have addressed my concerns with additional data, and I have no further comments or suggestions. This is an interesting study I am sure that it will be well cited.

Reviewer #2:

Remarks to the Author:

The revised version addressed most of my previous comments. The authors have done experiments to answer reviewers' critiques. I still feel that the evidence is not strong enough in demonstrating a high specificity for a general translation regulator toward JAK2 and then PD-L1 (without affecting other genes). I think the conclusion that "the RBP DENR is a novel regulator in PD-L1 expression" should be tuned down a bit.

Reviewer #3:

Remarks to the Author:

The authors have adequately addressed all of my concerns.

Reviewer #4:

Remarks to the Author:

The authors have addressed all my comments and I have no further requests.

Response to the Reviewer's comments point-by-point

Reviewer #1

The authors have addressed my concerns with additional data, and I have no further comments or suggestions. This is an interesting study I am sure that it will be well cited.

Thank you for your careful review. We really appreciate your efforts in reviewing our manuscript during this unprecedented and challenging time. We wish good health to you, your family, and community. Your careful review has helped to make our study clearer and more comprehensive.

Reviewer #2

The revised version addressed most of my previous comments. The authors have done experiments to answer reviewers' critiques. I still feel that the evidence is not strong enough in demonstrating a high specificity for a general translation regulator toward JAK2 and then PD-L1 (without affecting other genes). I think the conclusion that "the RBP DENR is a novel regulator in PD-L1 expression" should be tuned down a bit.

We thank the reviewer for this critical evaluation of our work. As suggested, we have rewritten the conclusion in Abstract "Overall, we discover an RBP DENR that could regulate PD-L1 expression for tumor immune evasion, and highlight the potential of DENR as a therapeutic target for immunotherapy." and removed the word "novel" in the Introduction.

We really appreciate your efforts in reviewing our manuscript during this unprecedented and challenging time. We wish good health to you, your family, and community. Your careful review has helped to make our study clearer and more comprehensive.

Reviewer #3

The authors have adequately addressed all of my concerns.

Thank you for your careful review. We really appreciate your efforts in reviewing our manuscript during this unprecedented and challenging time. We wish good health to you, your family, and community. Your careful review has helped to make our study clearer and more comprehensive.

Reviewer #4

The authors have addressed all my comments and I have no further requests.

Thank you for your careful review. We really appreciate your efforts in reviewing our manuscript during this unprecedented and challenging time. We wish good health to you, your family, and community. Your careful review has helped to make our study clearer and more comprehensive.